# Provably Convergent Policy Optimization via Metric-aware Trust Region Methods

**Jun Song**  *juns113@uw.edu*
*Department of Industrial and Systems Engineering*
*University of Washington*

**Niao He**  *niao.he@inf.ethz.ch*
*Department of Computer Science*
*ETH Zürich*

**Lijun Ding**  *lding47@wisc.edu*
*Wisconsin Institute for Discovery*
*University of Wisconsin - Madison*

**Chaoyue Zhao**  *cyzhao@uw.edu*
*Department of Industrial and Systems Engineering*
*University of Washington*

**Reviewed on OpenReview:** *https://openreview.net/forum?id=jkTqJJOGMS*

## Abstract

Trust-region methods based on Kullback-Leibler divergence are pervasively used to stabilize policy optimization in reinforcement learning. In this paper, we exploit more flexible metrics and examine two natural extensions of policy optimization with Wasserstein and Sinkhorn trust regions, namely *Wasserstein policy optimization (WPO)* and *Sinkhorn policy optimization (SPO)*. Instead of restricting the policy to a parametric distribution class, we directly optimize the policy distribution and derive their closed-form policy updates based on the Lagrangian duality. Theoretically, we show that WPO guarantees a monotonic performance improvement, and SPO provably converges to WPO as the entropic regularizer diminishes. Moreover, we prove that with a decaying Lagrangian multiplier to the trust region constraint, both methods converge to global optimality. Experiments across tabular domains, robotic locomotion, and continuous control tasks further demonstrate the performance improvement of both approaches, more robustness of WPO to sample insufficiency, and faster convergence of SPO, over state-of-art policy gradient methods.

## 1 Introduction

Policy-based reinforcement learning (RL) approaches have received remarkable success in many domains, including video games (Wang et al., 2017; Wu et al., 2017; Mnih et al., 2016), robotics (Grudic et al., 2003; Levine et al., 2016), and continuous control tasks (Duan et al., 2016; Schulman et al., 2016; Heess et al., 2015). One prominent example is policy gradient method (Grudic et al., 2003; Peters & Schaal, 2006; Lillicrap et al., 2016; Sutton et al., 1999; Williams, 1992; Mnih et al., 2016; Silver et al., 2014). The core idea is to represent the policy with a probability distribution $\pi_\theta(a|s) = P[a|s;\theta]$, such that the action $a$ in state $s$ is chosen stochastically following the policy $\pi_\theta$ controlled by parameter $\theta$. Determining the right step size to update the policy is crucial for maintaining the stability of policy gradient methods: too conservative choice of stepsizes result in slow convergence, while too large stepsizes may lead to catastrophically bad updates.

To control the size of policy updates, Kullback-Leibler (KL) divergence is commonly adopted to measure the difference between two policies. For example, the seminal work on trust region policy optimization (TRPO)

by Schulman et al. (2015) introduced KL divergence based constraints (trust region constraints) to restrict the size of the policy update; see also Peng et al. (2019); Abdolmaleki et al. (2018). Kakade (2001) and Schulman et al. (2017) introduced a KL-based penalty term to the objective to prevent excessive policy shift.

Though KL-based policy optimization has achieved promising results, it remains interesting whether using other metrics to gauge the similarity between policies could bring additional benefits. Recently, a few work (Richemond & Maginnis, 2017; Zhang et al., 2018; Moskovitz et al., 2021; Pacchiano et al., 2020) has explored the Wasserstein metric to restrict the deviation between consecutive policies. Compared with KL divergence, the Wasserstein metric has several desirable properties. Firstly, it is a true symmetric distance measure. Secondly, it allows flexible user-defined costs between actions and is less sensitive to ill-posed likelihood ratios. Thirdly but most importantly, the Wasserstein metric takes into account the geometry of the metric space (Panaretos & Zemel, 2019) and allows distributions to have different or even non-overlapping supports.

**Motivating Example**: Below we provide an example of a grid world (see Figure 1) that illustrates the advantages of using the Wasserstein metric over the KL divergence to construct trust regions and policy updates. The grid world consists of 5 regular grids and 2 goal grids, and there are three possible actions: left, right, and pickup. The player always starts from the middle grid, and making a left or right move results in a reward of $-1$. Picking up yields a reward of $-3$ at regular grids, $+5$ at the blue goal grid, and $+10$ at the red goal grid. An episode terminates either at the maximum length of 10 or immediately after picking up. We define the geometric distance between left and right actions to be 1, and 4 between other actions.

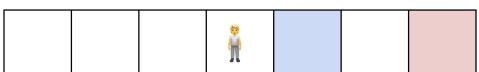

Figure 1: Motivating grid world example

Figure 2 shows the Wasserstein distance and KL divergence for different policy shifts of this grid world example. We can see that Wasserstein metric utilizes the geometric distance between actions to distinguish the shift of policy distribution to a close action (policy distribution $1 \rightarrow 2$ in Figure 2a) from the shift to a far action (policy distribution $1 \rightarrow 3$ in Figure 2b), while KL divergence does not. Figure 3 demonstrates the constrained policy updates based on Wasserstein distance and KL divergence respectively with a fixed trust region size 1. We can see that Wasserstein-based policy update finds the optimal policy faster than KL-based policy update. This is because KL distance is larger than Waserstein when considering policy shifts of close actions (see Figure 2a). Therefore, Wasserstein policy update is able to shift action (from left to right) in multiple states, while KL update is only allowed to shift action in a single state. Besides, KL policy update keeps using a suboptimal short-sighted solution between the 2nd and 4th iteration, which further slows down the convergence.

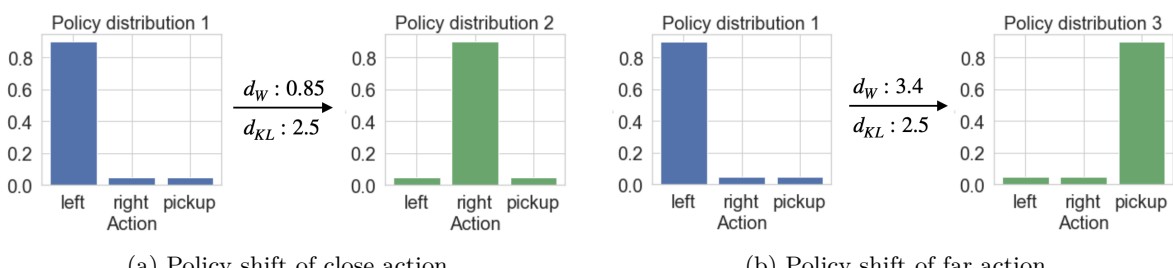

(a) Policy shift of close action        (b) Policy shift of far action

Figure 2: Wasserstein utilizes geometric feature of action space

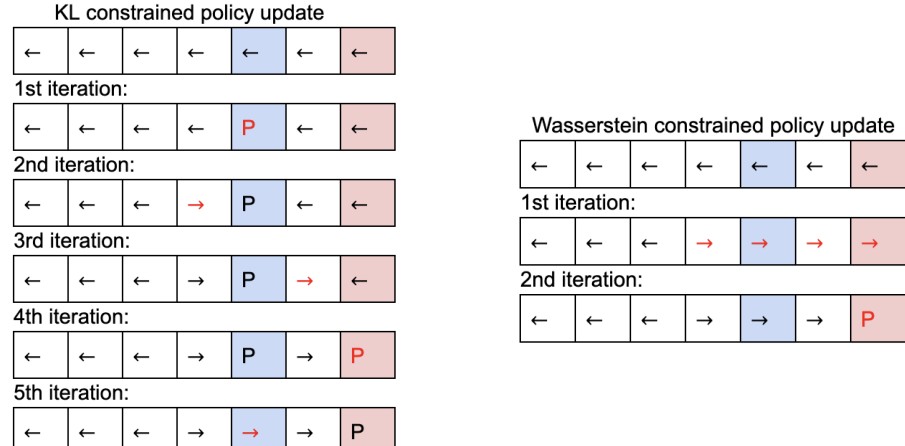

Figure 3: Demonstration of policy updates under different trust regions

However, the challenge of applying the Wasserstein metric for policy optimization is also evident: evaluating the Wasserstein distance requires solving an optimal transport problem, which could be computationally expensive. To avoid this computation hurdle, existing work resorts to different techniques to *approximate the policy update* under Wasserstein regularization. For example, Richemond & Maginnis (2017) solved the resulting RL problem using Fokker-Planck equations; Zhang et al. (2018) introduced particle approximation method to estimate the Wasserstein gradient flow. Recently, Moskovitz et al. (2021) instead considered the second-order Taylor expansion of Wasserstein distance based on Wasserstein information matrix to characterize the local behavioral structure of policies. Pacchiano et al. (2020) tackled behavior-guided policy optimization with smooth Wasserstein regularization by solving an approximate dual reformulation defined on reproducing kernel Hilbert spaces. Aside from such approximation, some of these work also limits the policy representation to a particular parametric distribution class, As indicated in Tessler et al. (2019), since parametric distributions are not convex in the distribution space, optimizing over such distributions results in local movements in the action space and thus leads to convergence to a sub-optimal solution. Until now, the theoretical performance of policy optimization under the Wasserstein metric remains elusive in light of these approximation errors.

In this paper, we study policy optimization with trust regions based on Wasserstein distance and Sinkhorn divergence. The latter is a smooth variant of Waserstein distance by imposing an entropic regularization to the optimal transport problem (Cuturi, 2013). We call them, *Wasserstein Policy Optimization (WPO)* and *Sinkhorn Policy Optimization (SPO)*, respectively. Instead of confining the distribution of policy to a particular distribution class, we work on the space of policy distribution directly, and consider all admissible policies that are within the trust regions with the goal of avoiding approximation errors. Unlike existing work, we focus on *exact characterization* of the policy updates. We would like to emphasize that our methodology and theoretical analysis in Section 3, 4, and 5 primarily concentrate on a discrete action space. However, we also present an extension of our method to accommodate a continuous action space, detailed in Section 7.5. We highlight our contributions as follows:

1. **Algorithms:** We develop closed-form expressions of the policy updates for both WPO and SPO based on the corresponding optimal Lagrangian multipliers of the trust region constraints. To the best of our knowledge, this is the first explicit closed-form updates for policy optimization based on Wasserstein and Sinkhorn trust regions. In particular, the optimal Lagrangian multiplier of SPO admits a simple form and can be computed efficiently. A practical on-policy actor-critic algorithm is proposed based on the derived expressions of policy updates and advantage value function estimation.

2. **Theory:** We theoretically show that WPO guarantees a *monotonic performance improvement* through the iterations, *even with non-optimal Lagrangian multipliers.* We also prove that SPO converges to WPO as the entropic regularizer diminishes. Moreover, we prove that with a decaying schedule of the multiplier,

SPO and WPO converge to *global optimality*, and with a constant multiplier, both methods converge *linearly* up to a neighborhood of the optimal value. To our best knowledge, this appears to be the first convergence rate analysis of policy optimization based on Wasserstein-type metrics.

3. **Experiments:** We provide comprehensive evaluation on the efficiency of WPO and SPO under several types of testing environments including tabular domains, robotic locomotion tasks, and further extend it to continuous control tasks. Compared to state-of-art policy gradients approaches that use KL divergence such as TRPO and PPO and those use Wasserstein metric such as Wasserstein Natural Policy Gradient (WNPG) (Moskovitz et al., 2021) and Behavior Guided Policy Gradients (BGPG) (Pacchiano et al., 2020), our methods achieve better sample efficiency, faster convergence, and improved final performance. Numerical study indicates that by properly choosing the weight of entropic regularizer, SPO achieves a better trade-off between convergence and final performance than WPO.

*Related work:* Wasserstein-like metrics have been explored in a number of works in the context reinforcement learning. Ferns et al. (2004) first introduced bisimulation metrics based on Wasserstein distance to quantify behavioral similarity between states for the purpose of state aggregation. Such bisimulation metrics were recently utilized for representation learning of RL; see e.g., Castro (2020); Agarwal et al. (2021b). In addition, a few recent work has also exploited Wasserstein distance for imitation learning (see e.g., Xiao et al. (2019); Dadashi et al. (2021)) and unsupervised RL (see e.g., He et al. (2022)). Our work is closely related to several previous studies, including Richemond & Maginnis (2017); Zhang et al. (2018); Moskovitz et al. (2021); Pacchiano et al. (2020), which also utilize Wasserstein distance to measure the proximity of policies. However, unlike the aforementioned studies that solely employ Wasserstein distance as an explicit penalty function, we additionally utilize it as a trust region constraint. Moreover, we consider nonparametric policies and derive explicit policy update forms, whereas these studies update parametric policies using policy gradients. Furthermore, we demonstrate monotonic performance improvement and global convergence with our policy update, which is not provided in these previous works. Regarding the use of Sinkhorn divergence in RL, Pacchiano et al. (2020)is the only related work to our best knowledge, where the entropy regularization is used to mitigate the computational burden of computing Wasserstein metric. However, no explicit form of policy update is provided in this work, while we derive an explicit Sinkhorn policy update and demonstrate its advantage in convergence speed. Additionally, we use Wasserstein distance to directly measure the proximity of nonparametric policies in the distribution space, while Pacchiano et al. (2020); Moskovitz et al. (2021)measure the similarity of parametric policies in the behavioral space.

Wasserstein-like metrics are also pervasively studied in distributionally robust optimization (DRO); see e.g., Esfahani & Kuhn (2018); Gao & Kleywegt (2016); Zhao & Guan (2018); Blanchet & Murthy (2019). We also point out that a recent concurrent work by Wang et al. (2021a) studied DRO using the Sinkhorn distance. Our duality formulations are largely inspired from existing work in DRO. However, we note that constrained policy optimization is conceptually different from DRO. Constrained policy optimization focuses on finding the optimistic policy that falls in a trust region, whereas DRO (e.g., the KL DRO) aims to optimize some worst-case loss given by the adversarial distribution of unknown parameters within some ambiguity set.

## 2 Background and Notations

**Markov Decision Process (MDP):** We consider an infinite-horizon discounted MDP, defined by the tuple $(\mathcal{S}, \mathcal{A}, P, r, \upsilon, \gamma)$, where $\mathcal{S}$ is the state space, $\mathcal{A}$ is the action space, $P : \mathcal{S} \times \mathcal{A} \times \mathcal{S} \to \mathbb{R}$ is the transition probability, $r : \mathcal{S} \times \mathcal{A} \to \mathbb{R}$ is the reward function, $\upsilon : \mathcal{S} \to \mathbb{R}$ is the distribution of the initial state $s_0$, and $\gamma \in (0, 1)$ is the discount factor. We define the return of timestep $t$ as the accumulated discounted reward from $t$, $R_t = \sum_{k=0}^{\infty} \gamma^k r(s_{t+k}, a_{t+k})$, and the value function as $V^\pi(s) = \mathbb{E}[R_t | s_t = s; \pi]$. The performance of a stochastic policy $\pi$ is defined as $J(\pi) = \mathbb{E}_{s_0, a_0, s_1 \dots}[\sum_{t=0}^{\infty} \gamma^t r(s_t, a_t)]$ where $a_t \sim \pi(a_t | s_t)$, $s_{t+1} \sim P(s_{t+1} | s_t, a_t)$. As shown in Kakade & Langford (2002), the expected return of a new policy $\pi'$ can be expressed in terms of the advantage over the old policy $\pi$: $J(\pi') = J(\pi) + \mathbb{E}_{s \sim \rho_\upsilon^{\pi'}, a \sim \pi'}[A^\pi(s, a)]$, where $A^\pi(s, a) = \mathbb{E}[R_t | s_t = s, a_t = a; \pi] - \mathbb{E}[R_t | s_t = s; \pi]$ represents the advantage function and $\rho_\upsilon^\pi$ represents the unnormalized discounted visitation frequencies with initial state distribution $\upsilon$, i.e., $\rho_\upsilon^\pi(s) = \mathbb{E}_{s_0 \sim \upsilon}[\sum_{t=0}^{\infty} \gamma^t P(s_t = s | s_0)]$.

**Trust Region Policy Optimization (TRPO):** In TRPO (Schulman et al., 2015), the policy $\pi$ is parameterized as $\pi_\theta$ with parameter vector $\theta$. For notation brevity, we use $\theta$ to represent the policy $\pi_\theta$. Then, the new policy $\theta'$ is found in each iteration to maximize the expected improvement $J(\pi') - J(\pi)$, or equivalently, the expected value of the advantage function:

$$\max_{\theta'} \quad \mathbb{E}_{s\sim\rho_v^\theta, a\sim\theta'}[A^\theta(s,a)]$$
$$\text{s.t.} \quad \mathbb{E}_{s\sim\rho_v^\theta}[d_{\text{KL}}(\theta', \theta)] \leq \delta,$$
(1)

where $d_{\text{KL}}$ represents the KL divergence and $\delta$ is the threshold of the distance between new and old policies.

**Wasserstein Distance:** Given two probability distributions of policies $\pi$ and $\pi'$ on the discrete action space $\mathcal{A} = \{a_1, a_2, \ldots, a_N\}$, the Wasserstein distance between the policies is defined as:

$$d_{\text{W}}(\pi', \pi) = \inf_{Q\in\Pi(\pi',\pi)} \langle Q, D\rangle,$$
(2)

where $\langle\cdot, \cdot\rangle$ denotes the Frobenius inner product. The infimum is taken over all joint distributions $Q$ with marginals $\pi'$ and $\pi$, and $D$ is the cost matrix with $D_{ij} = d(a_i, a_j)$, where $d(a_i, a_j)$ is defined as the distance between actions $a_i$ and $a_j$. Its largest entry in magnitude is denoted by $\|D\|_\infty$. In implementation, our choice of distance $d$ is task-dependent and is reported in Table 3 in Appendix A.

**Sinkhorn Divergence:** Sinkhorn divergence (Cuturi, 2013) provides a smooth approximation of the Wasserstein distance by adding an entropic regularizer. The Sinkhorn divergence is defined as:

$$d_{\text{S}}(\pi', \pi|\lambda) = \inf_{Q\in\Pi(\pi',\pi)} \left\{ \langle Q, D\rangle - \frac{1}{\lambda}h(Q) \right\},$$
(3)

where $h(Q) = -\sum_{i=1}^N \sum_{j=1}^N Q_{ij}\log Q_{ij}$ represents the entropy term, and $\lambda > 0$ is a regularization parameter. The intuition of adding the entropic regularization is: since most elements of the optimal joint distribution $Q$ will be 0 with a high probability, by trading the sparsity with entropy, a smoother and denser coupling between distributions can be achieved (Courty et al., 2014; 2016). Therefore, when the weight of the entropic regularization decreases (i.e., $\lambda$ increases), the sparsity of the divergence increases, and the Sinkhorn divergence converges to the Wasserstein metric, i.e., $\lim_{\lambda\to\infty} d_{\text{S}}(\pi', \pi|\lambda) = d_{\text{W}}(\pi', \pi)$. More critically, Sinkhorn divergence is useful to mitigate the computational burden of computing Wasserstein distance. In fact, the efficiency improvement that Sinkhorn divergence and the related algorithms brought paves the way to utilize Wasserstein-like metrics in many machine learning domains, including online learning (Cesa-Bianchi & Lugosi, 2006), model selection (Juditsky et al., 2008; Rigollet & Tsybakov, 2011), generative modeling (Genevay et al., 2018; Petzka et al., 2017; Patrini et al., 2019), dimensionality reduction (Huang et al., 2021; Lin et al., 2020; Wang et al., 2021b).

## 3 Wasserstein Policy Optimization

Motivated by TRPO, here we consider a trust region based on the Wasserstein metric. Moreover, we lift the restrictive assumption that a policy has to follow a parametric distribution class by allowing all admissible policies. Then, the new policy $\pi'$ is found in each iteration to maximize the estimated expected value of the advantage function. Therefore, the *Wasserstein Policy Optimization* (WPO) framework is shown as follows:

$$\max_{\pi'\in\mathcal{D}} \quad \mathbb{E}_{s\sim\rho_v^\pi, a\sim\pi'(\cdot|s)}[A^\pi(s,a)]$$
$$\text{where} \quad \mathcal{D} = \{\pi'|\mathbb{E}_{s\sim\rho_v^\pi}[d_{\text{W}}(\pi'(\cdot|s), \pi(\cdot|s))] \leq \delta\},$$
(4)

where the Wasserstein distance $d_{\text{W}}(\cdot, \cdot)$ is defined in (2).

In most practical cases, the reward $r$ is bounded and correspondingly, the accumulated discounted reward $R_t$ is bounded. So without loss of generality, we make the following assumption:

**Assumption 1.** *Assume $A^\pi(s,a)$ is bounded, i.e., $\sup_{a\in\mathcal{A}, s\in\mathcal{S}} |A^\pi(s,a)| \leq A^{max}$ for some $A^{max} > 0$.*

With Wasserstein metric based trust region constraint, we are able to derive the closed-form of the policy update shown in Theorem 1. The main idea is to form the Lagrangian dual of the constrained optimization problem presented above, which is inspired by the way to obtain the extremal distribution in Wasserstein DRO literature, see e.g., Kuhn et al. (2019); Blanchet & Murthy (2019); Zhao & Guan (2018). The detailed proof can be found in Appendix B.

**Theorem 1.** *(Closed-form policy update)* Let $\kappa_s^\pi(\beta, j) = argmax_{k=1\ldots N}\{A^\pi(s, a_k) - \beta D_{kj}\}$, where $D$ denotes the cost matrix. If Assumption 1 holds, then an optimal solution to (4) is:

$$\pi^*(a_i|s) = \sum_{j=1}^{N} \pi(a_j|s) f_s^*(i, j), \tag{5}$$

where $f_s^*(i, j) = 1$ if $i = \kappa_s^\pi(\beta^*, j)$ and $f_s^*(i, j) = 0$ otherwise, and $\beta^*$ is an optimal Lagrangian multiplier corresponds to the following dual formulation:

$$\min_{\beta \geq 0} F(\beta) = \min_{\beta \geq 0} \{\beta\delta + \mathbb{E}_{s\sim\rho_v^\pi}\sum_{j=1}^{N} \pi(a_j|s) \max_{i=1\ldots N}(A^\pi(s, a_i) - \beta D_{ij})\}. \tag{6}$$

*Moreover, we have $\beta^* \leq \bar{\beta}$, where $\bar{\beta} := \max_{s\in\mathcal{S}, k, j=1\ldots N, k\neq j} (D_{kj})^{-1}(A^\pi(s, a_k) - A^\pi(s, a_j))$.*

**Remark 1.** *For ease of notation and simplicity, we assume the uniqueness of $\kappa_s^\pi(\beta, j)$ in order to form the simple expression of $f_s^*$ in Theorem 1. When it is not unique, a necessary condition for the optimality of $\pi^*$ in (5) is $\sum_{i\in\mathcal{K}_s^\pi(\beta,j)} f_s^*(i, j) = 1$, and $f_s^*(i, j) = 0$ for $i \notin \mathcal{K}_s^\pi(\beta, j)$, where $\mathcal{K}_s^\pi(\beta, j) = argmax_{k=1\ldots N}A^\pi(s, a_k) - \beta D_{kj}$. The weight $f_s^*(i, j)$ for $i \in \mathcal{K}_s^\pi(\beta, j)$ could be determined through linear programming (see details in (17) in Appendix B).*

The exact policy update for WPO in (5) requires computing the optimal Lagrangian multiplier $\beta^*$ by solving the one-dimensional subproblem (6). A closed-form of $\beta^*$ is not easy to obtain in general, except for special cases of the distance $d(x, y)$ or cost matrix $D$. In Appendix C, we provide the closed-form of $\beta^*$ for the case when $d(x, y) = 0$ if $x = y$ and 1 otherwise.

**WPO Policy Update:** Based on Theorem 1, we introduce the following WPO policy updating rule:

$$\pi_{k+1}(a_i|s) = \mathbb{F}^{\mathrm{WPO}}(\pi_k) := \sum_{j=1}^{N} \pi_k(a_j|s) f_s^k(i, j), \tag{WPO}$$

where $f_s^k(i, j) = 1$ if $i = \kappa_s^{\pi_k}(\beta_k, j)$ and 0 otherwise. Note that different from (5), we allow $\beta_k$ to be chosen arbitrarily and time dependently. We show that such policy update always leads to a monotonic improvement of the performance even when $\beta_k$ is not the optimal Lagrangian multiplier. In particular, we propose two strategies to update multiplier $\beta_k$:

(i) Approximation of optimal $\beta_k$: To improve the convergence, we can approximately solve the optimal Lagrangian multiplier based on Sinkhorn divergence. More details in Section 4.

(ii) Time-dependent $\beta_k$: To improve the computational efficiency, we can simply treat $\beta_k$ as a time-dependent parameter, e.g., we can set $\beta_k$ as a diminishing sequence. In this setting, (WPO) produces the solution to the following penalty version of problem (4) (with $d = d_{\mathrm{W}}$):

$$\max_{\pi_{k+1}} \mathbb{E}_{s\sim\rho_v^{\pi_k}, a\sim\pi_{k+1}(\cdot|s)}[A^{\pi_k}(s, a)] - \beta_k\mathbb{E}_{s\sim\rho_v^{\pi_k}}[d(\pi_{k+1}(\cdot|s), \pi_k(\cdot|s))]. \tag{7}$$

## 4 Sinkhorn Policy Optimization

In this section, we introduce Sinkhorn policy optimization (SPO) by constructing trust region with Sinkhorn divergence. In the following theorem, we derive the optimal policy update in each step when using Sinkhorn divergence based trust region. Detailed proofs are provided in Appendix D.

**Theorem 2.** *If Assumption 1 holds, then the optimal solution to (4) with Sinkhorn divergence is:*

$$\pi_\lambda^*(a_i|s) = \sum_{j=1}^N \pi(a_j|s) f_{s,\lambda}^*(i,j), \tag{8}$$

*where $D$ denotes the cost matrix, $f_{s,\lambda}^*(i,j) = \dfrac{\exp\left(\frac{\lambda}{\beta_\lambda^*} A^\pi(s,a_i) - \lambda D_{ij}\right)}{\sum_{k=1}^N \exp\left(\frac{\lambda}{\beta_\lambda^*} A^\pi(s,a_k) - \lambda D_{kj}\right)}$ and $\beta_\lambda^*$ is an optimal solution to the following dual formulation:*

$$
\begin{aligned}
\min_{\beta \ge 0} F_\lambda(\beta) = \min_{\beta \ge 0} \Bigg\{ &\beta\delta - \mathbb{E}_{s \sim \rho_v^\pi} \sum_{j=1}^N \pi(a_j|s)\left(\frac{\beta}{\lambda} + \frac{\beta}{\lambda}\ln(\pi(a_j|s)) - \frac{\beta}{\lambda}\ln[\sum_{i=1}^N \exp\left(\frac{\lambda}{\beta} A^\pi(s,a_i) - \lambda D_{ij}\right)]\right) \\
&\mathbb{E}_{s \sim \rho_v^\pi} \sum_{i=1}^N \sum_{j=1}^N \frac{\beta}{\lambda} \frac{\exp\left(\frac{\lambda}{\beta} A^\pi(s,a_i) - \lambda D_{ij}\right) \cdot \pi(a_j|s)}{\sum_{k=1}^N \exp\left(\frac{\lambda}{\beta} A^\pi(s,a_k) - \lambda D_{kj}\right)} \Bigg\}.
\end{aligned}
\tag{9}
$$

*Moreover, we have $\beta_\lambda^* \le \frac{2A^{max}}{\delta}$.*

In contrast to the Wasserstein dual formulation (6), the objective in the Sinkhorn dual formulation (9) is differentiable in $\beta$ and admits closed-form gradients (shown in Appendix F). With this gradient information, we can use gradient-based global optimization algorithms (Wales & Doye, 1998; Zhan et al., 2006; Leary, 2000) to find a global optimal solution $\beta_\lambda^*$ to (9).

Next, we show that if the entropic regularization parameter $\lambda$ is large enough, then the optimal solution $\beta_\lambda^*$ is a close approximation to the $\beta^*$ of Wasserstein dual formulation. Proof is provided in Appendix G.

**Theorem 3.** *Define $\beta_{UB} = \max\{\frac{2A^{max}}{\delta}, \bar{\beta}\}$. We have:*

1. *$F_\lambda(\beta)$ converges to $F(\beta)$ uniformly on $[0, \beta_{UB}]$: $\lim_{\lambda \to \infty} \sup_{0 \le \beta \le \beta_{UB}} \left| F_\lambda(\beta) - F(\beta) \right| \le \lim_{\lambda \to \infty} \frac{\beta_{UB}}{\lambda} N \ln N = 0$.*

2. *$\lim_{\lambda \to \infty} argmin_{0 \le \beta \le \beta_{UB}} F_\lambda(\beta) \subseteq argmin_{0 \le \beta \le \beta_{UB}} F(\beta)$.*

Although it is difficult to obtain the exact value of the optimal solution $\beta^*$ to the Wasserstein dual formulation (6), the above theorem suggests that we can approximate $\beta^*$ via $\beta_\lambda^*$ by setting up a relative large $\lambda$. In practice, we can also adopt a smooth homotopy approach by setting an increasing sequence $\lambda_k$ for each iteration and letting $\lambda_k \to \infty$.

**SPO Policy Update:**  Based on Theorem 2, we introduce the following SPO policy updating rule:

$$\pi_{k+1}(a_i|s) = \mathbb{F}^{\text{SPO}}(\pi_k) = \sum_{j=1}^N \pi_k(a_j|s) f_{s,\lambda_k}^k(i,j). \tag{SPO}$$

Here $f_{s,\lambda_k}^k(i,j) = \dfrac{\exp\left(\frac{\lambda_k}{\beta_k} A^{\pi_k}(s,a_i) - \lambda_k D_{ij}\right)}{\sum_{l=1}^N \exp\left(\frac{\lambda_k}{\beta_k} A^{\pi_k}(s,a_l) - \lambda_k D_{lj}\right)}$, $\lambda_k \ge 0$ and $\beta_k \ge 0$ are some control parameters. The parameter $\beta_k$ can be either computed via solving the one-dimensional subproblem (9) or simply set as a diminishing sequence. The proper setup of $\lambda_k$ can effectively adjust the trade-off between convergence speed and final performance. More details are provided in the ablation study in Section 7.

## 5    Theoretical Analysis

We first show that SPO policy update converges to WPO policy update as the regularization parameter increases (i.e., $\lambda \to \infty$). The detailed proof is provided in Appendix H.

**Lemma 1.** *As $\lambda_k \to \infty$, SPO update converges to WPO update: $\lim_{\lambda_k \to \infty} \mathbb{F}^{SPO}(\pi_k) \in \mathbb{F}^{WPO}(\pi_k)$.*

We then provide a theoretical justification that WPO policy update (and SPO with $\lambda \to \infty$) are always guaranteed to improve the true performance $J$ monotonically if we have access to the true advantage function. If the advantage function can only be evaluated inexactly with limited samples, then an extra estimation error (measured by the largest absolute entry $\|\cdot\|_\infty$) will occur. Proof can be found in Appendix I.

**Theorem 4.** *(Performance improvement) For any initial state distribution $\upsilon$ and any $\beta_k \geq 0$, if $\|\hat{A}^\pi - A^\pi\|_\infty \leq \epsilon$ for some $\epsilon > 0$, let $\hat{\mathcal{K}}_s^{\pi_k}(\beta_k, j) = argmax_{i=1,\dots,N}\{\hat{A}^{\pi_k}(s, a_i) - \beta_k D_{ij}\}$, WPO policy update (and SPO with $\lambda \to \infty$) guarantee the following performance improvement bound when the inaccurate advantage function $\hat{A}^\pi$ is used,*

$$J(\pi_{k+1}) \geq J(\pi_k) + \beta_k \mathbb{E}_{s \sim \rho_\upsilon^{\pi_{k+1}}} \sum_{j=1}^N \pi_k(a_j|s) \sum_{i \in \hat{\mathcal{K}}_s^{\pi_k}(\beta_k, j)} f_s^k(i, j) D_{ij} - \frac{2\epsilon}{1 - \gamma}. \tag{10}$$

The value of $\epsilon$, which quantifies the approximation error of the advantage function, is dependent on various factors such as the advantage estimation algorithm used and the number of samples (Schulman et al., 2016). It is worth noting that the improvement bound of NPG/TRPO (Cen et al., 2021)includes the same additional term $-\frac{2\epsilon}{1-\gamma}$, which indicates that our methods offer comparable theoretical performance guarantees to KL based updates. In the following, we show that with a decreasing schedule of the multiplier $\beta_k$, both WPO and SPO policy updates have their values $J(\pi_k)$ converging to the optimal $J^\star = \max_\pi J(\pi)$ on the tabular domain. To start, for $k$-th iteration, we consider (WPO) and (SPO) (with arbitrary $\lambda > 0$) whose updates $\pi_{k+1}$ are optimal solutions to (7) with $d$ being $d_W$ and $d_S$ respectively.

**Assumption 2.** *The state space and the action space are both finite, the reward function $r$ is non-negative, and the initial distribution covers all state.*

Note that once state and action spaces are both finite, the reward can be assumed non-negative without loss of generality, as we can always add $\max_{s,a} |r(s, a)|$ to the reward function without changing the optimal policy and the order of the policies. Defining the optimal value function $V^\star(s) = \max_\pi \mathbb{E}[R_t|s_t = s]$, we have the following theorem, whose proof is in Appendix J and is inspired by Bhandari & Russo (2021).

**Theorem 5.** *(Global convergence) Under Assumption 2, we have for any $\beta_k \geq 0$, (WPO) satisfies that*

$$\|V^\star - V^{\pi_{k+1}}\|_\infty \leq \gamma \|V^\star - V^{\pi_k}\|_\infty + \beta_k \|D\|_\infty, \tag{11}$$

*and (SPO) satisfies that*

$$\|V^\star - V^{\pi_{k+1}}\|_\infty \leq \gamma \|V^\star - V^{\pi_k}\|_\infty + 2\frac{\beta_k}{1 - \gamma} \left( \|D\|_\infty + 2\frac{\log N}{\lambda} \right). \tag{12}$$

*If $\lim_{k \to \infty} \beta_k = 0$, we further have $\lim_{k \to \infty} J(\pi_k) = J^\star$.*

**Remark 2.** *Note the convergence is* geometric. *If we keep $\beta_k$ as a constant, then $0 \leq J^\star - J(\pi^T) \leq \|V^\star - V^{\pi^T}\|_\infty \leq \gamma^T \|V^\star - V^{\pi_0}\|_\infty + \frac{\beta B}{1 - \gamma}$, where $B = \|D\|_\infty$ for (WPO) and $B = 2\frac{\|D\|_\infty + 2\frac{\log N}{\lambda}}{1 - \gamma}$ for (SPO). To achieve an $\epsilon$ optimality gap, we only need to take $\beta = \frac{(1 - \gamma)\epsilon}{2B}$ and let $T \geq \frac{\log(\epsilon/2)}{\gamma}$.*

**Remark 3.** *The study of global non-asymptotic convergence of nonconvex policy optimization algorithms has been an active research topic. Recent theoretical work has mostly centered on PG and natural policy gradient (NPG) Kakade (2001) - a close relative of TRPO; see e.g., Agarwal et al. (2021a); Cen et al. (2021); Lan (2022). To our best knowledge, a few work has discussed the global convergence of TRPO. Neu et al. (2017) and Geist et al. (2019) established the connection of TRPO to Mirror Descent, but did not provide any non-asymptotic rate; Shani et al. (2020) showed that adaptive TRPO with decaying stepsize achieved $O(1/\sqrt{T})$ convergence rate for unregularized MDPs in the tabular setting (finite state and finite action). Our result seems to be the first non-asymptotic analysis of policy optimization based on Wasserstein and Sinkhorn divergence. It remains interesting to extend the convergence theory of TRPO/WPO/SPO to function approximation regime following recent advance Agarwal et al. (2021a). However, this is beyond the scope of our current work, as we focus on explicit closed-form update of WPO/SPO, which can be a viable alternative to TRPO in practice.*

## 6 A Practical Algorithm

In practice, the advantage value functions are often estimated from sampled trajectories. In this section, we provide a practical on-policy actor-critic algorithm, described in Algorithm 1, that combines WPO/SPO with advantage function estimation.

At each iteration, the first step is to collect trajectories, which can be either complete or partial. If the trajectory is complete, the total return can be directly expressed as the accumulated discounted rewards $R_t = \sum_{k=0}^{T-t-1} \gamma^k r_{t+k}$. If the trajectory is partial, it can be estimated by applying the multi-step temporal difference (TD) methods (De Asis et al., 2017): $\hat{R}_{t:t+n} = \sum_{k=0}^{n-1} \gamma^k r_{t+k} + \gamma^n V(s_{t+n})$. Then for the advantage estimation, we can use Monte Carlo advantage estimation, i.e., $\hat{A}_t^{\pi_k} = R_t - V_{\psi_k}(s_t)$ or Generalized Advantage Estimation (GAE) (Schulman et al., 2016), which provides a more explicit control over the bias-variance trade-off. In the value update step, we use a neural net to represent the value function, where $\psi$ is the parameter that specifies the value net $s \to V(s)$. Then, we can update $\psi$ by using gradient descent, which significantly reduces the computational burden of computing advantage directly. The computational complexity of the algorithm is discussed in Appendix K.

---

**Algorithm 1:** On-policy WPO/SPO algorithm

---

Input: number of iterations $K$, learning rate $\alpha$
Initialize policy $\pi_0$ and value network $V_{\psi_0}$ with random parameter $\psi_0$
**for** $k = 0, 1, 2 \ldots K$ **do**
    Collect trajectory set $\mathcal{D}_k$ on policy $\pi_k$
    For each timestep $t$ in each trajectory, compute total returns $G_t$ and estimate advantages $\hat{A}_t^{\pi_k}$
    Update value:
    $\psi_{k+1} \leftarrow \psi_k - \alpha \nabla_{\psi_k} \sum (G_t - V_{\psi_k}(s_t))^2$
    Update policy:
    $\pi_{k+1} \leftarrow \mathbb{F}(\pi_k)$ via WPO/ SPO with $\hat{A}_t^{\pi_k}$
**end**

---

## 7 Experiments

In this section, we evaluate the proposed WPO and SPO approaches presented in Algorithm 1. We compare the performance of our methods with benchmarks including TRPO (Schulman et al., 2015), PPO (Schulman et al., 2017), A2C (Mnih et al., 2016); and with BGPG (Pacchiano et al., 2020), WNPG (Moskovitz et al., 2021) for continuous control. The code of our WPO/SPO can be found here[1]. We adopt the implementations of TRPO, PPO and A2C from OpenAI Baselines (Dhariwal et al., 2017) for MuJuCo tasks and Stable Baselines (Hill et al., 2018) for other tasks. For BGPG, we adopt the same implementation[2] as (Pacchiano et al., 2020).

Our experiments include (1) ablation study that focuses on sensitivity analysis of WPO and SPO; (2) tabular domain tasks with discrete state and action including the Taxi, Chain, and Cliff Walking environments; (3) locomotion tasks with continuous state and discrete action including the CartPole, Acrobot environments; (4) comparison of KL and Wasserstein trust regions under tabular domain and locomotion tasks; and (5) extension to continuous control tasks with continuous action including HalfCheetah, Hopper, Walker, and Ant environments from MuJuCo. See Table 4 in Appendix A for a summary of performance. The setting of hyperparameters and network sizes of our algorithms and additional results are provided in Appendix A.

### 7.1 Ablation Study

In this experiment, we first examine the sensitivity of WPO in terms of different strategies of $\beta_k$. We test four settings of $\beta$ value for WPO policy update: (1) Setting 1: Computing optimal $\beta$ value for all policy update; (2) Setting 2: Computing optimal $\beta$ value for first 20% of policy updates and decaying $\beta$ for the

---

[1]https://github.com/efficientwpo/EfficientWPO
[2]https://github.com/behaviorguidedRL/BGRL

remaining; (3) Setting 3: Computing optimal $\beta$ value for first 20% of policy updates and fix $\beta$ as its last updated value for the remaining; (4) Setting 4: Decaying $\beta$ for all policy updates (e.g., $\beta_k = \Theta(1/\log k)$). In particular, Setting 2 is rooted in the observation that $\beta^*$ decays slowly in the later stage of the experiments carried out in the paper. Small perturbations are added to the approximate values to avoid any stagnation in updating. Taxi task (Dieterich, 1998) from tabular domain is selected for this experiment.

Table 1: Runtime for different $\beta$ settings, average across 5 runs with random initialization

| Runtime | Taxi (s) | CartPole (s) |
|---|---|---|
| Setting 1 (optimal $\beta$) | $1224.3 \pm 105.7$ | $129.7 \pm 15.2$ |
| Setting 2 (optimal-then-decay) | $648.4 \pm 55.7$ | $63.2 \pm 8.3$ |
| Setting 3 (optimal-then-fix) | $630.2 \pm 67.4$ | $67.1 \pm 9.7$ |
| Setting 4 (decaying $\beta$) | $522.7 \pm 49.5$ | $44.3 \pm 6.2$ |

The performance comparisons and average run times are shown in Figure 4 and Table 1 respectively. Figure 4a and Table 1 clearly indicate a tradeoff between computation efficiency and accuracy in terms of different choices of $\beta$ value. Setting 2 is the most effective way to balance the tradeoff between performance and run time. For the rest of experiments, we adopt this setting for both WPO and SPO (see Appendix A.2 for how Setting 2 is tuned for each task). Figure 4b shows that as $\lambda$ increases, the convergence of SPO becomes slower but the final performance of SPO improves and becomes closer to that of WPO, which verifies the convergence property of Sinkhorn to Wasserstein distance shown in Theorem 3. Therefore, the choice of $\lambda$ can effectively adjust the trade-off between convergence and final performance. Similar results are observed when using time-varying $\lambda$ on Taxi, Chain and CartPole tasks, presented in Figure 9 in Appendix A.

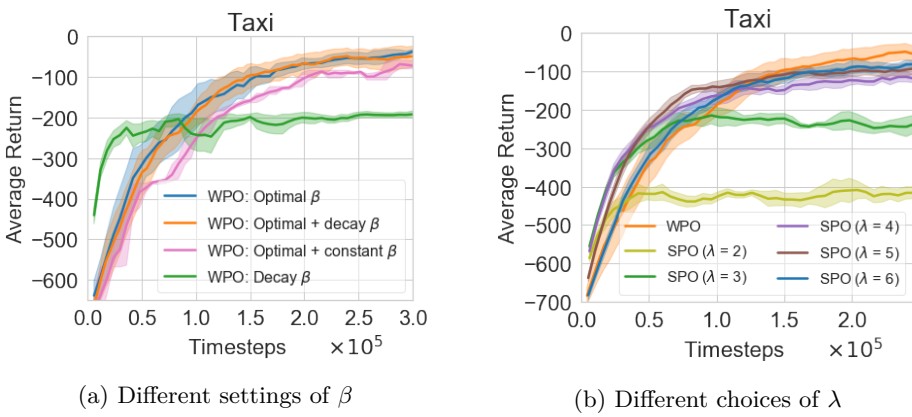

(a) Different settings of $\beta$        (b) Different choices of $\lambda$

Figure 4: Episode rewards for Taxi with different $\beta$ and $\lambda$ settings, averaged across 5 runs with random initialization. The shaded area depicts the mean $\pm$ the standard deviation.

## 7.2   Tabular Domains

We evaluate WPO and SPO on tabular domain tasks and test the exploration ability of the algorithms on several environments including Taxi, Chain, and Cliff Walking. We use a table of size $|\mathcal{S}| \times |\mathcal{A}|$ to represent the policy $\pi(a|s)$. For the value function, we use a neural net to smoothly update the values. The performance of WPO and SPO are compared to the performance of TRPO, PPO and A2C under the same neural net structure. Results on Taxi, Cliff and Chain are reported in Figure 5.

As shown in Figure 5, the performances of WPO, SPO and TRPO are manifestly better than A2C and PPO. Among the trust region based methods, WPO and SPO outperform TRPO in Taxi and Cliff Walking, whereas in Chain, the performances of these three methods are comparable. In all of the test cases, SPO converges faster than WPO but to a lower optimum. As further shown in Table 2, for the Taxi environment, WPO has a higher successful drop-off rate and a lower task completion time while the original TRPO reaches the time limit with a drop-off rate 0, suggesting that WPO finds a better policy than the original TRPO. In Figure 7,

we also compare the performance of WPO under Wasserstein and KL divergences given different number of samples $N_A$ used to estimate the advantage function, and the result suggests that using Wasserstein metric is more robust than KL divergence under inaccurate advantage values.

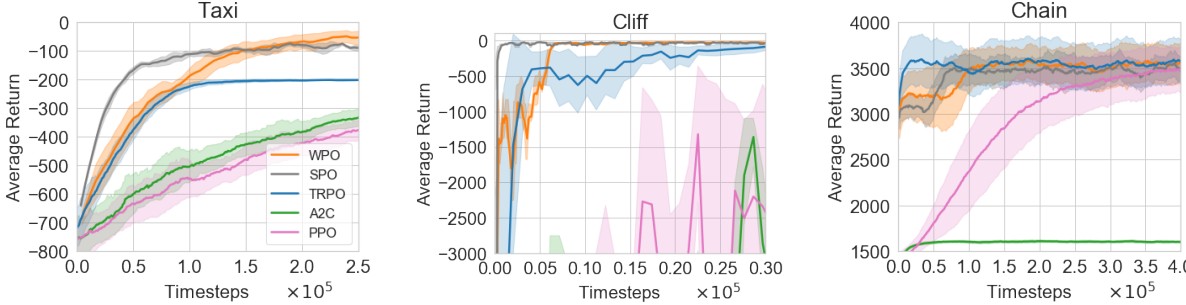

Figure 5: Episode rewards during training for tabular domain tasks, averaged across 5 runs with random initialization. The shaded area depicts the mean ± the standard deviation.

Table 2: Trained agents performance on Taxi

|      | Success (+20) | Fail (-10) | Steps (-1) | Return   |
|------|---------------|------------|------------|----------|
| WPO  | 0.753         | 0.232      | 70.891     | -58.151  |
| TRPO | 0             | 0          | 200        | -200     |

## 7.3    Robotic Locomotion Tasks

We now integrate deep neural network architecture into WPO and SPO and evaluate their performance on several locomotion tasks (with continuous state and discrete action), including CartPole (Barto et al., 1983) and Acrobot (Geramifard et al., 2015). We use two separate neural nets to represent the policy and the value. The policy neural net receives state $s$ as an input and outputs the categorical distribution of $\pi(a|s)$. A random subset of states $\mathcal{S}_k \in \mathcal{S}$ is sampled at each iteration to perform policy updates.

Figure 6 shows that WPO and SPO outperform TRPO, PPO and A2C in most tasks in terms of final performance, except in Acrobot where PPO performs the best. In most cases, SPO converges faster but WPO has a better final performance. To train $10^5$ timesteps in the discrete locomotion tasks, the training wall-clock time is 63.2 ±8.2$s$ for WPO, 64.7 ±7.8$s$ for SPO, 59.4 ±10.2$s$ for TRPO and 69.9 ±10.5$s$ for PPO. Therefore, WPO has a similar computational efficiency as TRPO and PPO.

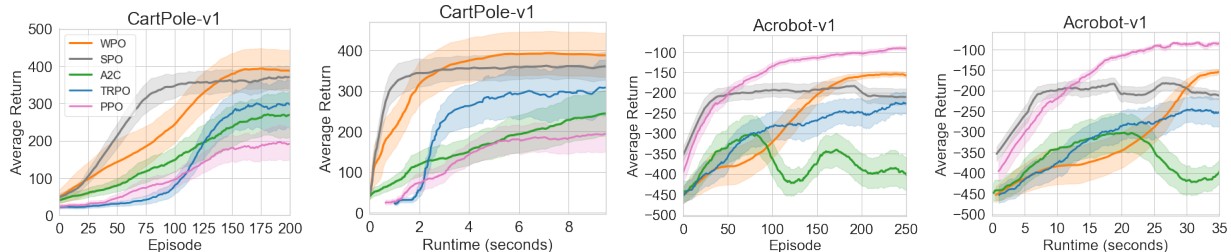

Figure 6: Episode rewards during the training process for the locomotion tasks, averaged across 5 runs with random initialization. The shaded area depicts the mean ± the standard deviation.

## 7.4    Comparison of Wasserstein and KL Trust Regions

We show that compared with the KL divergence, the utilization of Wasserstein metric can cope with the inaccurate advantage estimations caused by the lack of samples. Let $N_A$ denote the number of samples used

to estimate the advantage function. We evaluate the performance of WPO framework (4) with Wasserstein and KL constraints (as derived in Peng et al. (2019)). We consider the Chain task and different $N_A$. As shown in Figure 7, when $N_A$ is 1000, KL performs slightly better than WPO. However, when $N_A$ decreases to 100 or 250, WPO outperforms KL. These results indicate that WPO is more robust than KL under inaccurate advantage values. This finding is consistent with our observations on the policy update formulations of Wasserstein and KL. For the Wasserstein update in (5), policy will be updated only when the advantage difference between two actions is significant, i.e., $A^\pi(s, a_j) - \beta D_{ij} \geq A^\pi(s, a_i)$. However, for the KL update in Peng et al. (2019), policy will be updated as long as the current advantage function has a single non-zero value. Therefore, KL update is more sensitive; while Wasserstein update is more robust and more tolerant to advantage inaccuracies. Similar results are obtained for the locomotion tasks (Figure 10 in Appendix A). The runtime of Wasserstein and KL updates are reported in Table 5 in Appendix A.

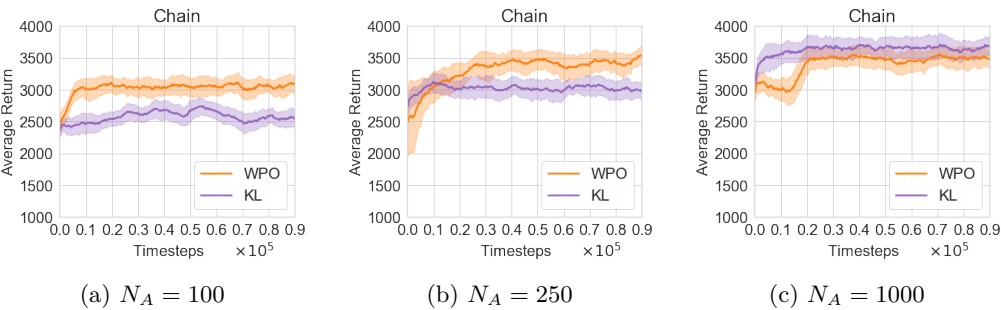

(a) $N_A = 100$             (b) $N_A = 250$             (c) $N_A = 1000$

Figure 7: Episode rewards during training for the Chain task, where advantage value function is estimated under different number of samples, averaged across 5 runs with random initialization. The shaded area depicts the mean $\pm$ the standard deviation.

### 7.5 Extension to Continuous Control

To extend to environments with continuous action, we use Implicit Quantile Networks (IQN) (Will Dabney & Munos, 2018) actor that can represent an arbitrary complex non-parametric policy. Let $F_s^{-1}(p)$ represent the quantile function associated with policy $\pi(\cdot|s)$. The IQN actor takes state $s$ and probability $p \in [0, 1]$ as input, and outputs the corresponding quantile value $a = F_s^{-1}(p)$. IQN actor can be trained to approach pre-defined target policy distributions through quantile regression (Will Dabney & Munos, 2018; Tessler et al., 2019).

Define the action support for state $s$ in $k$-th iteration as $I^{\pi_k}(s) = \{a' : A^{\pi_k}(s, a') > \min_{a \in I^{\pi_{k-1}}(s)} A^{\pi_k}(s, a)\}$. Then, the WPO/SPO target policy distribution to guide IQN update in the $k$-th iteration is:

$$P_{I^{\pi_k}(s)}(a'|s) = \sum_{a \in I^{\pi_{k-1}}(s)} \pi_k(a|s) f_s(a', a), \tag{13}$$

where for WPO update $f_s(a', a) = 1$ if $a' = \mathrm{argmax}_{a' \in I^{\pi_k}(s)}\{A^{\pi_k}(s, a') - \beta_k d(a', a)\}$ and $f_s(a', a) = 0$ otherwise; for SPO update, $f_s(a', a) = \frac{\exp(\frac{\lambda_k}{\beta_k} A^{\pi_k}(s, a') - \lambda_k d(a', a))}{\sum_{a' \in I^{\pi_k}(s)} \exp(\frac{\lambda_k}{\beta_k} A^{\pi_k}(s, a') - \lambda_k d(a', a))}$. In implementation, we sample a batch of states $\mathcal{S}_k \in \mathcal{S}$ at each iteration to perform policy updates, and for each $s \in \mathcal{S}_k$, we sample $|\mathcal{A}_k|$ actions to approximate the support $I^{\pi_k}(s)$ and the target policy distribution $P_{I^{\pi_k}(s)}(\cdot|s)$.

We additionally compare WPO and SPO with BGPG (Pacchiano et al., 2020) and WNPG (Moskovitz et al., 2021) that are specially designed to address the continuous control with Wasserstein metric, for several MuJuCo tasks including HalfCheetah, Hopper, Walker, and Ant. Figure 8 shows that WPO and SPO have consistently better performances than other benchmarks. Similar results are obtained for the challenging Humanoid task, presented in Figure 11 in Appendix A. We also provide the runtime of each algorithm in Table 6 in Appendix A.

Figure 8: Episode rewards during training for MuJuCo continuous control tasks, averaged across 10 runs with random initialization. The shaded area depicts the mean $\pm$ the standard deviation.

## 8 Conclusion

In this paper, we present two policy optimization frameworks, WPO and SPO, which can exactly characterize the policy updates instead of confining their distributions to particular distribution class or requiring any approximation. Our methods outperform TRPO and PPO with better sample efficiency, faster convergence, and improved final performance. Our numerical results show that the Wasserstein metric is more robust to the ambiguity of advantage functions, compared with the KL divergence. Our strategy for adjusting $\beta$ value for WPO can reduce the computational time and boost the convergence without noticeable performance degradation. SPO improves the convergence speed of WPO by properly choosing the weight of the entropic regularizer. Performance improvement and global convergence for WPO are discussed. For future work, it remains interesting to extend the idea to PPO and natural policy gradients, which penalize the policy update instead of imposing trust region constraint, and extend it to off-policy frameworks.

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

# A    Implementation Details and Additional Results

The implementation of our WPO/SPO can be found in https://github.com/efficientwpo/EfficientWPO. We use the implementations of TRPO, PPO and A2C from OpenAI Baselines (Dhariwal et al., 2017) for MuJuCo tasks and Stable Baselines (Hill et al., 2018) for other tasks. For BGPG, we adopt the same implementation as Pacchinao et al., (2020) based on the released code https://github.com/behaviorguidedRL/BGRL.

## A.1    Visitation Frequencies Estimation:

The unnormalized discounted visitation frequencies are needed to compute the global optimal $\beta^*$. At the $k$-th iteration, the visitation frequencies $\rho_k^\pi$ are estimated using samples of the trajectory set $\mathcal{D}_k$. Specifically, we first initialize $\rho_k^\pi(s) = 0$, $\forall s \in S$. Then for each timestep $t$ in each trajectory from $\mathcal{D}_k$, we update $\rho_k^\pi$ as $\rho_k^\pi(s_t) \leftarrow \rho_k^\pi(s_t) + \gamma^t/|\mathcal{D}_k|$.

## A.2    Optimal-then-decay Beta Strategy:

During the training of multiple tasks, including Taxi, Chain and CartPole, we observe a consistent trend in the behavior of the optimal $\beta$ value during the policy updates: It initially fluctuates, then stabilizes and decays slowly towards 0. In the Taxi task, the optimal $\beta$ stabilizes after approximately 18% of the total training iterations. If we decay $\beta$ before this stabilization point (e.g, using optimal beta for only first 5% or 10% updates), we observe a drop in performance. However, we do not observe any notable performance difference when we decay $\beta$ after this stabilization point (e.g., using optimal $\beta$ for first 20% or 30% updates). We also observe that the optimal $\beta$ decays at a very slow rate, and $\Theta(1/\log(k))$ matches this trend best. If we employ a faster decaying function, such as $\Theta(1/k)$ or $\Theta(1/k^2)$, we observe a drop in performance.

Based on these findings, when implementing the optimal-then-decay $\beta$ strategy on other tasks, we compute the optimal $\beta$ for each policy update until we observe that its value stabilizes across updates. At this point, we stop calculating the optimal $\beta$ and decay it using $\Theta(1/\log(k))$ for the remaining policy updates. The specific iteration at which the optimal $\beta$ value stabilizes varies across tasks, and we denote this point as $k_\beta$, which is reported in Table 3.

## A.3    Hyperparameters and Performance Summary

Our main experimental results are reported in section 7. In addition, we provide the setting of hyperparameters and network sizes of our WPO/SPO algorithms in Table 3, and a summary of performance in Table 4.

Table 3: Hyperparameters and network sizes

| | Taxi-v3 | NChain-v0 CliffWalking-v0 | CartPole-v1 | Acrobot-v1 | MuJuCo tasks |
|---|---|---|---|---|---|
| $\gamma$ | 0.9 | 0.9 | 0.95 | 0.95 | 0.99 |
| $lr_\pi$ | \ | \ | $10^{-2}$ | $5 \times 10^{-3}$ | $10^{-4}$ |
| $lr_{\text{value}}$ | $10^{-2}$ | $10^{-2}$ | $10^{-2}$ | $5 \times 10^{-3}$ | $10^{-3}$ |
| $|\mathcal{D}_k|$ | 60 (Taxi) | 1 (Chain) 3 (CliffWalking) | 2 | 3 | partial |
| $\pi$ size | 2D array | 2D array | $[64, 64]$ | $[64, 64]$ | $[400, 300]$ |
| Q/v size | $[10, 7, 5]$ | $[10, 7, 5]$ | $[64, 64]$ | $[64, 64]$ | $[400, 300]$ |
| $|\mathcal{S}_k|$ | all states, $|\mathcal{S}|$ | all states, $|\mathcal{S}|$ | 128 | 128 | 64 |
| $|\mathcal{A}_k|$ | all actions, $|\mathcal{A}|$ | all actions, $|\mathcal{A}|$ | all actions, $|\mathcal{A}|$ | all actions, $|\mathcal{A}|$ | 32 |

| $d(a, a')$ | 0-1 distance [3] | 0-1 distance | 0-1 distance | 0-1 distance | L1 distance |
|---|---|---|---|---|---|
| $k_\beta$ | 250 | 100 (Chain) | 150 | 150 | 1000 |
| | | 50 (CliffWalking) | | | |

Table 4: Averaged rewards over last 10% episodes during the training process

| Environment | WPO | SPO | TRPO | PPO | A2C | BGPG | WNPG |
|---|---|---|---|---|---|---|---|
| Taxi-v3 | $-45 \pm 27$ | $-87 \pm 11$ | $-202 \pm 3$ | $-381 \pm 34$ | $-338 \pm 30$ | - | - |
| NChain-v0 | $3549 \pm 197$ | $3432 \pm 131$ | $3522 \pm 258$ | $3506 \pm 237$ | $1606 \pm 10$ | - | - |
| CliffWalking-v0 | $-35 \pm 15$ | $-25 \pm 1$ | $-159 \pm 94$ | $-3290 \pm 2106$ | $-5587 \pm 1942$ | - | - |
| CartPole-v1 | $388 \pm 54$ | $370 \pm 30$ | $297 \pm 65$ | $193 \pm 45$ | $267 \pm 61$ | - | - |
| Acrobot-v1 | $-162 \pm 8$ | $-185 \pm 15$ | $-248 \pm 33$ | $-103 \pm 5$ | $-379 \pm 39$ | - | - |
| HalfCheetah-v2 | $2050 \pm 108$ | $1750 \pm 172$ | $1158 \pm 35$ | $1628 \pm 136$ | $-645 \pm 31$ | $1697 \pm 195$ | $1832 \pm 125$ |
| Hopper-v2 | $3208 \pm 259$ | $2834 \pm 305$ | $2035 \pm 248$ | $2321 \pm 233$ | $43 \pm 21$ | $1982 \pm 218$ | $2361 \pm 272$ |
| Walker2d-v2 | $3739 \pm 298$ | $3489 \pm 257$ | $2535 \pm 369$ | $3290 \pm 354$ | $28 \pm 1$ | $2775 \pm 301$ | $3059 \pm 209$ |
| Ant-v2 | $1863 \pm 271$ | $1780 \pm 257$ | $21 \pm 10$ | $1487 \pm 206$ | $-39 \pm 8$ | $1622 \pm 235$ | $1587 \pm 221$ |
| Humanoid-v2 | $965 \pm 76$ | $914 \pm 93$ | $725 \pm 112$ | $632 \pm 73$ | $107 \pm 15$ | $797 \pm 85$ | $820 \pm 91$ |

## A.4 Additional Results for Ablation Studies

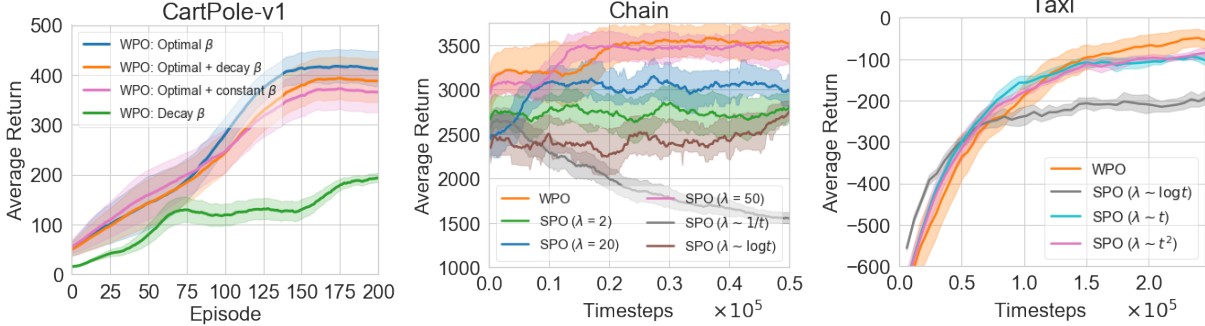

Figure 9: Episode rewards during the training process for different $\beta$ and $\lambda$ settings, averaged across 5 runs with a random initialization. The shaded area depicts the mean $\pm$ the standard deviation.

---

[3]We note that specifying distance based on control relevance leads to higher performance in this test case: i.e., $d = 1$ to distinct actions from set $A = \{$ move north, move south, move west, move east $\}$, $d = 1$ to distinct actions from set $B = \{$ pickup, dropoff $\}$, and $d = 4$ to actions from different sets.

## A.5 Additional Comparison of Wasserstein and KL Trust Regions

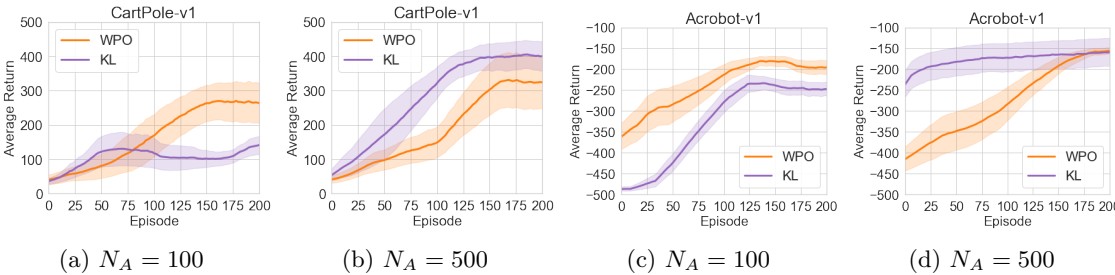

(a) $N_A = 100$     (b) $N_A = 500$     (c) $N_A = 100$     (d) $N_A = 500$

Figure 10: Episode rewards during the training process for the locomotion tasks, averaged across 5 runs with a random initialization. The shaded area depicts the mean $\pm$ the standard deviation.

Table 5: Average runtime (seconds) of WPO, SPO and KL

|  | WPO | SPO | KL |
|---|---|---|---|
| Taxi-v3 (per $10^3$ steps) | $71.0 \pm 7.3$ | $69.5 \pm 8.7$ | $74.3 \pm 9.5$ |
| NChain-v0 (per $10^3$ steps) | $58.4 \pm 9.1$ | $63.1 \pm 7.4$ | $59.9 \pm 8.7$ |
| CartPole-v1 (per $10^6$ steps) | $11.4 \pm 1.8$ | $10.2 \pm 2.3$ | $9.7 \pm 1.9$ |
| Acrobot-v1 (per $10^5$ steps) | $10.4 \pm 1.9$ | $9.7 \pm 2.5$ | $10.9 \pm 2.3$ |
| Humanoid-v2 (per $10^5$ steps) | $422.7 \pm 65.4$ | $409.1 \pm 46.5$ | $438.5 \pm 61.2$ |

## A.6 Additional Results for Large-scale Continuous Control

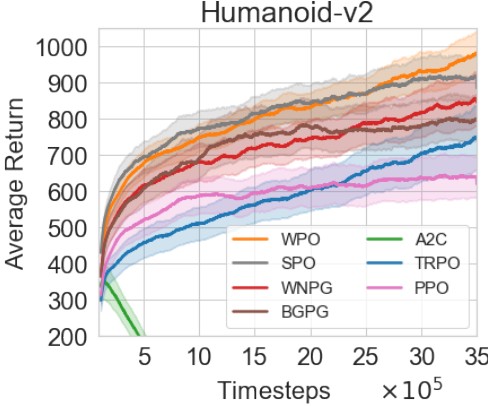

Figure 11: Episode rewards during training for MuJuCo Humanoid task, averaged across 10 runs with random initialization. The shaded area depicts the mean $\pm$ the standard deviation.

Table 6: Average runtime (seconds per $10^5$ timesteps) for the MuJuCo continuous control tasks

| Environment | WPO | SPO | TRPO | PPO | A2C | BGPG | WNPG |
|---|---|---|---|---|---|---|---|
| HalfCheetah-v2 | $297 \pm 31$ | $289 \pm 25$ | $290 \pm 28$ | $292 \pm 36$ | $293 \pm 27$ | $306 \pm 33$ | $298 \pm 22$ |
| Hopper-v2 | $233 \pm 38$ | $226 \pm 42$ | $242 \pm 56$ | $167 \pm 36$ | $254 \pm 49$ | $201 \pm 32$ | $197 \pm 31$ |
| Walker2d-v2 | $289 \pm 55$ | $312 \pm 61$ | $253 \pm 39$ | $307 \pm 52$ | $259 \pm 46$ | $322 \pm 62$ | $214 \pm 45$ |
| Ant-v2 | $307 \pm 51$ | $290 \pm 57$ | $296 \pm 63$ | $251 \pm 47$ | $291 \pm 41$ | $286 \pm 63$ | $269 \pm 54$ |
| Humanoid-v2 | $423 \pm 65$ | $401 \pm 47$ | $446 \pm 52$ | $395 \pm 57$ | $230 \pm 31$ | $425 \pm 58$ | $398 \pm 49$ |

# B  Proof of Theorem 1

**Theorem 1.** *(Closed-form policy update)* *Let* $\kappa_s^\pi(\beta, j) = argmax_{k=1...N}\{A^\pi(s, a_k) - \beta D_{kj}\}$, *where $D$ denotes the cost matrix. If Assumption 1 holds, then an optimal solution to (4) is:*

$$\pi^*(a_i|s) = \sum_{j=1}^{N} \pi(a_j|s) f_s^*(i, j), \tag{5}$$

*where $f_s^*(i, j) = 1$ if $i = \kappa_s^\pi(\beta^*, j)$ and $f_s^*(i, j) = 0$ otherwise, and $\beta^*$ is an optimal Lagrangian multiplier corresponds to the following dual formulation:*

$$\min_{\beta \geq 0} F(\beta) = \min_{\beta \geq 0} \{\beta\delta + \mathbb{E}_{s \sim \rho_v^\pi} \sum_{j=1}^{N} \pi(a_j|s) \max_{i=1...N} (A^\pi(s, a_i) - \beta D_{ij})\}. \tag{6}$$

*Moreover, we have $\beta^* \leq \bar{\beta}$, where $\bar{\beta} := \max_{s \in \mathcal{S}, k, j=1...N, k \neq j} (D_{kj})^{-1}(A^\pi(s, a_k) - A^\pi(s, a_j))$.*

*Proof of Theorem 1.* First, we denote $Q^s$ as the joint distribution of $\pi(\cdot|s)$ and $\pi'(\cdot|s)$ with $\sum_{i=1}^{N} Q_{ij}^s = \pi(a_j|s)$ and $\sum_{j=1}^{N} Q_{ij}^s = \pi'(a_i|s)$. Also, let $f_s(i, j)$ represent the conditional distribution of $\pi'(a_i|s)$ under $\pi(a_j|s)$. Then $Q_{ij}^s = \pi(a_j|s) f_s(i, j)$, $\pi'(a_i|s) = \sum_{j=1}^{N} Q_{ij}^s = \sum_{j=1}^{N} \pi(a_j|s) f_s(i, j)$. In addition:

$$d_W(\pi'(\cdot|s), \pi(\cdot|s)) = \min_{Q_{ij}^s} \sum_{i=1}^{N} \sum_{j=1}^{N} D_{ij} Q_{ij}^s = \min_{f_s(i,j)} \sum_{i=1}^{N} \sum_{j=1}^{N} D_{ij} \pi(a_j|s) f_s(i, j), \text{ and}$$

$$\mathbb{E}_{a \sim \pi'(\cdot|s)}[A^\pi(s, a)] = \sum_{i=1}^{N} A^\pi(s, a_i) \pi'(a_i|s) = \sum_{i=1}^{N} \sum_{j=1}^{N} A^\pi(s, a_i) \pi(a_j|s) f_s(i, j).$$

Thus, the WPO problem in (4) can be reformulated as:

$$\max_{f_s(i,j) \geq 0} \mathbb{E}_{s \sim \rho_v^\pi} \sum_{i=1}^{N} \sum_{j=1}^{N} A^\pi(s, a_i) \pi(a_j|s) f_s(i, j) \tag{14a}$$

$$s.t. \quad \mathbb{E}_{s \sim \rho_v^\pi} \sum_{i=1}^{N} \sum_{j=1}^{N} D_{ij} \pi(a_j|s) f_s(i, j) \leq \delta, \tag{14b}$$

$$\sum_{i=1}^{N} f_s(i, j) = 1, \qquad \forall s \in \mathcal{S}, j = 1 \ldots N. \tag{14c}$$

Note here that (14b) is equivalent to $\mathbb{E}_{s \sim \rho_v^\pi} \min_{f_s(i,j)} \sum_{i=1}^{N} \sum_{j=1}^{N} D_{ij} \pi(a_j|s) f_s(i, j) \leq \delta$ because if we have a feasible $f_s(i, j)$ to make (14b) hold, we must have $\mathbb{E}_{s \sim \rho_v^\pi} \min_{f_s(i,j)} \sum_{i=1}^{N} \sum_{j=1}^{N} D_{ij} \pi(a_j|s) f_s(i, j) \leq \delta$.

Since both the objective function and the constraint are linear in $f_s(i, j)$, (14) is a convex optimization problem. Also, Slater's condition holds for (14) as the feasible region has an interior point, which is $f_s(i, i) = 1$ $\forall i$, and $f_s(i, j) = 0$ $\forall i \neq j$. Meanwhile, since $A^\pi(s, a)$ is bounded based on Assumption 1, the objective is bounded above. Therefore, strong duality holds for (14). At this point we can derive the dual problem of (14) as its equivalent reformulation:

$$\min_{\beta \geq 0, \zeta_j^s} \beta\delta + \int_{s \in \mathcal{S}} \sum_{j=1}^{N} \zeta_j^s ds$$

$$s.t. \quad A^\pi(s, a_i) \pi(a_j|s) - \beta D_{ij} \pi(a_j|s) - \frac{\zeta_j^s}{\rho_v^\pi(s)} \leq 0, \qquad \forall s \in \mathcal{S}, i, j = 1 \ldots N. \tag{15}$$

We observe that with a fixed $\beta$, the optimal $\zeta_j^s$ will be achieved at:

$$\zeta_j^{s*}(\beta) = \max_{i=1...N} \rho_v^\pi(s) \pi(a_j|s) (A^\pi(s, a_i) - \beta D_{ij}). \tag{16}$$

Denote $\beta^*$ as an optimal solution to (15) and $f_s^*(i,j)$ as an optimal solution to (14). Due to the complimentary slackness, the following equations hold:

$$(A^\pi(s,a_i)\pi(a_j|s) - \beta^* D_{ij}\pi(a_j|s) - \frac{\zeta_j^{s*}(\beta^*)}{\rho_v^\pi(s)})f_s^*(i,j) = 0, \qquad \forall s,i,j.$$

In this case, $f_s^*(i,j)$ can have non-zero values only when $A^\pi(s,a_i)\pi(a_j|s) - \beta^* D_{ij}\pi(a_j|s) - \frac{\zeta_j^{s*}(\beta^*)}{\rho_v^\pi(s)} = 0$, which means $\zeta_j^{s*}(\beta^*) = \rho_v^\pi(s)\pi(a_j|s)(A^\pi(s,a_i) - \beta^* D_{ij})$. Given the expression of the optimal $\zeta_j^{s*}$ in (16), $f_s^*(i,j)$ can have non-zero values only when $i \in \mathcal{K}_s^\pi(\beta^*,j)$, where $\mathcal{K}_s^\pi(\beta,j) = \text{argmax}_{k=1...N} A^\pi(s,a_k) - \beta D_{kj}$.

When there exists a unique optimizer, i.e., $|\mathcal{K}_s^\pi(\beta^*,j)| = 1$, let $\kappa_s^\pi(\beta^*,j)$ denote the optimizer. Since $\sum_{i=1}^N f_s^*(i,j) = 1$ as indicated in (14c), the only optimal solution is:

$$f_s^*(i,j) = \begin{cases} 1 & \text{if } i = \kappa_s^\pi(\beta^*,j), \\ 0 & \text{otherwise.} \end{cases}$$

When there exists multiple optimizers, i.e., $|\mathcal{K}_s^\pi(\beta^*,j)| > 1$, the optimal weights $f_s^*(i,j)$ for $i \in \mathcal{K}_s^\pi(\beta^*,j)$ could be determined by solving the following linear programming:

$$\max_{f_s^*(i,j)\geq 0, i\in\mathcal{K}_s^\pi(\beta^*,j)} \quad \mathbb{E}_{s\sim\rho_v^\pi} \sum_{j=1}^N \pi(a_j|s) \sum_{i\in\mathcal{K}_s^\pi(\beta^*,j)} A^\pi(s,a_i) f_s^*(i,j)$$

$$s.t. \quad \mathbb{E}_{s\sim\rho_v^\pi} \sum_{j=1}^N \pi(a_j|s) \sum_{i\in\mathcal{K}_s^\pi(\beta^*,j)} D_{ij} f_s^*(i,j) \leq \delta, \tag{17}$$

$$\sum_{i\in\mathcal{K}_s^\pi(\beta^*,j)} f_s^*(i,j) = 1, \qquad \forall s\in\mathcal{S}, j=1\ldots N.$$

And then the corresponding optimal solution is, $\pi^*(a_i|s) = \sum_{j=1}^N \pi(a_j|s)f_s^*(i,j)$.

Last, by substituting $\zeta_j^{s*}(\beta) = \rho_v^\pi(s)\pi(a_j|s)\max_{i=1...N}(A^\pi(s,a_i) - \beta D_{ij})$ into the dual problem (15), we can reformulate (15) into:

$$\min_{\beta\geq 0}\{\beta\delta + \int_{s\in\mathcal{S}} \sum_{j=1}^N \zeta_j^{s*}(\beta)ds\} = \min_{\beta\geq 0}\{\beta\delta + \mathbb{E}_{s\sim\rho_v^\pi} \sum_{j=1}^N \pi(a_j|s) \max_{i=1...N}(A^\pi(s,a_i) - \beta D_{ij})\}. \tag{18}$$

The optimal $\beta$ can then be obtained by solving (18).

We will further show that $\beta^* \leq \bar{\beta} := \max_{s\in\mathcal{S},k,j=1...N,k\neq j} (D_{kj})^{-1}(A^\pi(s,a_k) - A^\pi(s,a_j))$.

In the general case, i.e., $\beta \geq 0$, (14a) is non-negative because:

$$\mathbb{E}_{s\sim\rho_v^\pi} \sum_{i=1}^N \sum_{j=1}^N A^\pi(s,a_i)\pi(a_j|s)f_s^*(i,j) \tag{19a}$$

$$= \mathbb{E}_{s\sim\rho_v^\pi} \sum_{j=1}^N \pi(a_j|s) \sum_{i=1}^N A^\pi(s,a_i)f_s^*(i,j) \tag{19b}$$

$$= \mathbb{E}_{s\sim\rho_v^\pi} \sum_{j=1}^N \pi(a_j|s) \sum_{i\in\mathcal{K}_s^\pi(\beta^*,j)} f_s^*(i,j)A^\pi(s,a_i) \tag{19c}$$

$$\geq \mathbb{E}_{s\sim\rho_v^\pi} \sum_{j=1}^N \pi(a_j|s) \sum_{i\in\mathcal{K}_s^\pi(\beta^*,j)} f_s^*(i,j)[A^\pi(s,a_j) + \beta^* D_{ij}] \tag{19d}$$

$$= \mathbb{E}_{s\sim\rho_v^\pi} \sum_{j=1}^{N} \pi(a_j|s) A^\pi(s,a_j) + \mathbb{E}_{s\sim\rho_v^\pi} \sum_{j=1}^{N} \pi(a_j|s) \sum_{i\in\mathcal{K}_s^\pi(\beta^*,j)} f_s^*(i,j)\beta^* D_{ij} \tag{19e}$$

$$= \mathbb{E}_{s\sim\rho_v^\pi} \sum_{j=1}^{N} \pi(a_j|s)\beta^* \sum_{i\in\mathcal{K}_s^\pi(\beta^*,j)} f_s^*(i,j) D_{ij} \tag{19f}$$

$$\geq 0, \tag{19g}$$

where (19d) holds since for $i \in \mathcal{K}_s^\pi(\beta^*,j)$, $A^\pi(s,a_i) - \beta^* D_{ij} \geq A^\pi(s,a_j) - \beta^* D_{jj} = A^\pi(s,a_j)$. When $\beta^* > \max_{s\in\mathcal{S},k,j=1...N,k\neq j}\{\frac{A^\pi(s,a_k)-A^\pi(s,a_j)}{D_{kj}}\}$, we have that for all $s \in \mathcal{S}$, $\kappa_s^\pi(\beta^*,j) = j$. Thus, $f_s^*(i,i) = 1$, $\forall i$ and $f_s^*(i,j) = 0$, $\forall i \neq j$. The objective value (14a) will be 0 because $\mathbb{E}_{s\sim\rho_v^\pi} \sum_{i=1}^{N}\sum_{j=1}^{N} A^\pi(s,a_i)\pi(a_j|s)f_s^*(i,j) = \mathbb{E}_{s\sim\rho_v^\pi} \sum_{i=1}^{N} A^\pi(s,a_i)\pi(a_i|s) = 0$. The left hand side of (14b) equals to $\mathbb{E}_{s\sim\rho_v^\pi} \sum_{i=1}^{N}\sum_{j=1}^{N} D_{ij}\pi(a_j|s)f_s^*(i,j) = \mathbb{E}_{s\sim\rho_v^\pi} \sum_{i=1}^{N} D_{ii}\pi(a_i|s) = 0$. Thus, for any $\delta > 0$, (14b) is always satisfied.

Since the objective of the primal Wasserstein trust-region constrained problem in (6) constantly evaluates to 0 when $\beta^* > \max_{s\in\mathcal{S},k,j=1...N,k\neq j}\{\frac{A^\pi(s,a_k)-A^\pi(s,a_j)}{D_{kj}}\}$, and is non-negative when $\beta^* \leq \max_{s\in\mathcal{S},k,j=1...N,k\neq j}\{\frac{A^\pi(s,a_k)-A^\pi(s,a_j)}{D_{kj}}\}$, we can use $\max_{s\in\mathcal{S},k,j=1...N,k\neq j}\{\frac{A^\pi(s,a_k)-A^\pi(s,a_j)}{D_{kj}}\}$ as an upper bound for the optimal dual variable $\beta^*$.

$\square$

## C  Optimal Beta for a Special Distance

**Proposition 1.** *Let $k_s = argmax_{i=1,...,N} A^\pi(s,a_i)$, we have:*

*(1). If the initial point $\beta_0$ is in $[\max_{s,j}\{A^\pi(s,a_{k_s}) - A^\pi(s,a_j)\}, +\infty)$, the local optimal $\beta$ solution is $\max_{s,j}\{A^\pi(s,a_{k_s}) - A^\pi(s,a_j)\}$.*
*(2). If the initial point $\beta_0$ is in $[0, \min_{s,j\neq k_s}\{A^\pi(s,a_{k_s}) - A^\pi(s,a_j)\}]$: if $\delta - \int_{s\in\mathcal{S}} \rho^\pi(s)(1 - \pi(a_{k_s}|s))ds < 0$, the local optimal $\beta$ is $\min_{s,j\neq k_s}\{A^\pi(s,a_{k_s}) - A^\pi(s,a_j)\}$; otherwise, the local optimal $\beta$ solution is 0.*

*(3). If the initial point $\beta_0$ is in $(\min_{s,j\neq k_s}\{A^\pi(s,a_{k_s}) - A^\pi(s,a_j)\}, \max_{s,j}\{A^\pi(s,a_{k_s}) - A^\pi(s,a_j)\})$, we construct sets $I_s^1$ and $I_s^2$ as:*

*$\mathbf{for}\ s \in \mathcal{S}, j \in \{1,2...N\}$ : $\mathbf{if}\ \beta_0 \geq A^\pi(s,a_{k_s}) - A^\pi(s,a_j)$ $\mathbf{then}$ Add $j$ to $I_s^1$ $\mathbf{else}$ Add $j$ to $I_s^2$. Then, if $\delta - \mathbb{E}_{s\sim\rho^\pi} \sum_{j\in I_s^2} \pi(a_j|s) < 0$, the local optimal $\beta$ is $\min_{s\in\mathcal{S},j\in I_s^2}\{A^\pi(s,a_{k_s}) - A^\pi(s,a_j)\}$; otherwise, the local optimal $\beta$ is $\max_{s\in\mathcal{S},j\in I_s^1}\{A^\pi(s,a_{k_s}) - A^\pi(s,a_j)\}$.*

*Proof of Proposition 1.* (1). When $\beta \in [\max_{s,j}\{A^\pi(s,a_{k_s}) - A^\pi(s,a_j)\}, +\infty)$, we have $A^\pi(s,a_j) \geq A^\pi(s,a_{k_s}) - \beta$ for all $s \in \mathcal{S}$, $j = 1...N$. Since $A^\pi(s,a_{k_s}) - \beta \geq A^\pi(s,a_k) - \beta$ for all $k = 1...N$, we have $A^\pi(s,a_j) \geq A^\pi(s,a_k) - \beta$ for all $s \in \mathcal{S}$, $j = 1...N$, $k = 1...N$. Thus, $j \in argmax_{k=1...N}\{A^\pi(s,a_k) - \beta D_{kj}\}$, for all $s \in \mathcal{S}$, $j = 1...N$. Therefore, (6) can be reformulated as:

$$\min_{\beta\geq 0}\{\beta\delta + \mathbb{E}_{s\sim\rho_v^\pi} \sum_{j=1}^{N} \pi(a_j|s) A^\pi(s,a_j)\}.$$

Since $\delta \geq 0$, we have the local optimal $\beta = \max_{s,j}\{A^\pi(s,a_{k_s}) - A^\pi(s,a_j)\}$.

(2). When $\beta \in [0, \min_{s,j\neq k_s}\{A^\pi(s,a_{k_s}) - A^\pi(s,a_j)\}]$, we have $A^\pi(s,a_j) \leq A^\pi(s,a_{k_s}) - \beta$ for all $s \in \mathcal{S}$, $j = 1...N$, $j \neq k_s$. Thus $k_s \in argmax_{k=1...N}\{A^\pi(s,a_k) - \beta D_{kj}\}$ for all $s \in \mathcal{S}$, $j = 1...N$. The inner part of

(6) then is:

$$\beta\delta + \mathbb{E}_{s\sim\rho_v^\pi}\{\sum_{j=1,j\neq k_s}^N \pi(a_j|s)(A^\pi(s,a_{k_s}) - \beta) + \pi(a_{k_s}|s)A^\pi(s,a_{k_s})\}$$

$$= \beta(\delta - \mathbb{E}_{s\sim\rho_v^\pi}\sum_{j=1,j\neq k_s}^N \pi(a_j|s)) + \mathbb{E}_{s\sim\rho_v^\pi}\sum_{j=1}^N \pi(a_j|s)A^\pi(s,a_{k_s})$$

$$= \beta(\delta - \int_{s\in\mathcal{S}}\rho_v^\pi(s)(1 - \pi(a_{k_s}|s))ds) + \mathbb{E}_{s\sim\rho_v^\pi}\sum_{j=1}^N \pi(a_j|s)A^\pi(s,a_{k_s}).$$

If $\delta - \int_{s\in\mathcal{S}}\rho_v^\pi(s)(1 - \pi(a_{k_s}|s))ds < 0$, we have the local optimal $\beta = \min_{s,j\neq k_s}\{A^\pi(s,a_{k_s}) - A^\pi(s,a_j)\}$. If $\delta - \int_{s\in\mathcal{S}}\rho_v^\pi(s)(1 - \pi(a_{k_s}|s))ds \geq 0$, we have the local optimal $\beta = 0$.

(3). For an initial point $\beta_0$ in $(\min_{s,j\neq k_s}\{A^\pi(s,a_{k_s}) - A^\pi(s,a_j)\}, \max_{s,j}\{A^\pi(s,a_{k_s}) - A^\pi(s,a_j)\})$, we construct partitions $I_s^1$ and $I_s^2$ of the set $\{1, 2 \ldots N\}$ in the way described in Proposition 1 for all $s \in \mathcal{S}$. Consider $\beta$ in the neighborhood of $\beta_0$, i.e., $\beta \geq A^\pi(s,a_{k_s}) - A^\pi(s,a_j)$ for $s \in \mathcal{S}, j \in I_s^1$ and $\beta \leq A^\pi(s,a_{k_s}) - A^\pi(s,a_j)$ for $s \in \mathcal{S}, j \in I_s^2$. Then the inner part of (6) can be reformulated as:

$$\beta\delta + \mathbb{E}_{s\sim\rho_v^\pi}\{\sum_{j\in I_s^1}\pi(a_j|s)A^\pi(s,a_j) + \sum_{j\in I_s^2}\pi(a_j|s)(A^\pi(s,a_{k_s}) - \beta)\}$$

$$= \beta(\delta - \mathbb{E}_{s\sim\rho_v^\pi}\sum_{j\in I_s^2}\pi(a_j|s)) + \mathbb{E}_{s\sim\rho_v^\pi}\{\sum_{j\in I_s^1}\pi(a_j|s)A^\pi(s,a_j) + \sum_{j\in I_s^2}\pi(a_j|s)A^\pi(s,a_{k_s})\}.$$

If $\delta - \mathbb{E}_{s\sim\rho_v^\pi}\sum_{j\in I_s^2}\pi(a_j|s) < 0$, we have the local optimal $\beta = \min_{s\in\mathcal{S},j\in I_s^2}\{A^\pi(s,a_{k_s}) - A^\pi(s,a_j)\}$. If $\delta - \mathbb{E}_{s\sim\rho_v^\pi}\sum_{j\in I_s^2}\pi(a_j|s) \geq 0$, we have the local optimal $\beta = \max_{s\in\mathcal{S},j\in I_s^1}\{A^\pi(s,a_{k_s}) - A^\pi(s,a_j)\}$. $\square$

## D    Proof of Theorem 2

**Theorem 2.** *If Assumption 1 holds, then the optimal solution to (4) with Sinkhorn divergence is:*

$$\pi_\lambda^*(a_i|s) = \sum_{j=1}^N \pi(a_j|s)f_{s,\lambda}^*(i,j), \tag{8}$$

*where $D$ denotes the cost matrix, $f_{s,\lambda}^*(i,j) = \frac{\exp\left(\frac{\lambda}{\beta_\lambda^*}A^\pi(s,a_i) - \lambda D_{ij}\right)}{\sum_{k=1}^N \exp\left(\frac{\lambda}{\beta_\lambda^*}A^\pi(s,a_k) - \lambda D_{kj}\right)}$ and $\beta_\lambda^*$ is an optimal solution to the following dual formulation:*

$$\min_{\beta\geq 0} F_\lambda(\beta) = \min_{\beta\geq 0}\left\{\beta\delta - \mathbb{E}_{s\sim\rho_v^\pi}\sum_{j=1}^N \pi(a_j|s)(\frac{\beta}{\lambda} + \frac{\beta}{\lambda}\ln(\pi(a_j|s)) - \frac{\beta}{\lambda}\ln[\sum_{i=1}^N \exp\left(\frac{\lambda}{\beta}A^\pi(s,a_i) - \lambda D_{ij}\right)])\right.$$

$$\left.\mathbb{E}_{s\sim\rho_v^\pi}\sum_{i=1}^N\sum_{j=1}^N \frac{\beta}{\lambda}\frac{\exp\left(\frac{\lambda}{\beta}A^\pi(s,a_i) - \lambda D_{ij}\right)\cdot\pi(a_j|s)}{\sum_{k=1}^N \exp\left(\frac{\lambda}{\beta}A^\pi(s,a_k) - \lambda D_{kj}\right)}\right\}. \tag{9}$$

*Moreover, we have $\beta_\lambda^* \leq \frac{2A^{max}}{\delta}$.*

*Proof of Theorem 2.* Invoking the definition of Sinkhorn divergence in (3), the trust region constrained problem with Sinkhorn divergence can be reformulated as:

$$\max_{Q} \quad \mathbb{E}_{s\sim\rho_v^\pi}[\sum_{i=1}^N A^\pi(s,a_i)\sum_{j=1}^N Q_{ij}^s] \tag{20a}$$

$$s.t. \quad \mathbb{E}_{s\sim\rho_v^\pi}[\sum_{i=1}^N\sum_{j=1}^N D_{ij}Q_{ij}^s + \frac{1}{\lambda}Q_{ij}^s\log Q_{ij}^s] \leq \delta \tag{20b}$$

$$\sum_{i=1}^{N} Q_{ij}^s = \pi(a_j|s), \quad \forall j = 1, \ldots, N, s \in \mathcal{S}. \tag{20c}$$

Let $\beta$ and $\omega$ represent the dual variables of constraints (20b) and (20c) respectively, then the Lagrangian duality of (20) can be derived as:

$$\max_{Q} \min_{\beta \geq 0, \omega} L(Q, \beta, \omega) = \max_{Q} \min_{\beta \geq 0, \omega} \mathbb{E}_{s \sim \rho_v^\pi} [\sum_{i=1}^{N} A^\pi(s, a_i) \sum_{j=1}^{N} Q_{ij}^s]$$

$$+ \int_{s \in \mathcal{S}} \sum_{j=1}^{N} \omega_j^s (\sum_{i=1}^{N} Q_{ij}^s - \pi(a_j|s)) ds + \beta(\delta - \mathbb{E}_{s \sim \rho_v^\pi} [\sum_{i=1}^{N} \sum_{j=1}^{N} D_{ij} Q_{ij}^s + \frac{1}{\lambda} Q_{ij}^s \log Q_{ij}^s]) \tag{21a}$$

$$= \max_{Q} \min_{\beta \geq 0, \omega} \mathbb{E}_{s \sim \rho_v^\pi} [\sum_{i=1}^{N} A^\pi(s, a_i) \sum_{j=1}^{N} Q_{ij}^s] + \int_{s \in \mathcal{S}} \sum_{j=1}^{N} \sum_{i=1}^{N} \frac{\omega_j^s}{\rho_v^\pi(s)} Q_{ij}^s \rho_v^\pi(s) ds$$

$$- \int_{s \in \mathcal{S}} \sum_{j=1}^{N} \omega_j^s \pi(a_j|s) ds + \beta\delta - \beta \mathbb{E}_{s \sim \rho_v^\pi} [\sum_{i=1}^{N} \sum_{j=1}^{N} D_{ij} Q_{ij}^s + \frac{1}{\lambda} Q_{ij}^s \log Q_{ij}^s]) \tag{21b}$$

$$= \max_{Q} \min_{\beta \geq 0, \omega} \beta\delta - \int_{s \in \mathcal{S}} \sum_{j=1}^{N} \omega_j^s \pi(a_j|s) ds$$

$$+ \mathbb{E}_{s \sim \rho_v^\pi} [\sum_{i=1}^{N} \sum_{j=1}^{N} (A^\pi(s, a_i) - \beta D_{ij} + \frac{\omega_j^s}{\rho_v^\pi(s)}) Q_{ij}^s - \frac{\beta}{\lambda} Q_{ij}^s \log Q_{ij}^s] \tag{21c}$$

$$= \min_{\beta \geq 0, \omega} \max_{Q} \beta\delta - \int_{s \in \mathcal{S}} \sum_{j=1}^{N} \omega_j^s \pi(a_j|s) ds$$

$$+ \mathbb{E}_{s \sim \rho_v^\pi} [\sum_{i=1}^{N} \sum_{j=1}^{N} (A^\pi(s, a_i) - \beta D_{ij} + \frac{\omega_j^s}{\rho_v^\pi(s)}) Q_{ij}^s - \frac{\beta}{\lambda} Q_{ij}^s \log Q_{ij}^s], \tag{21d}$$

where (21d) holds since the Lagrangian function $L(Q, \beta, \omega)$ is concave in $Q$ and linear in $\beta$ and $\omega$, and we can exchange the max and the min following the Minimax theorem (Sion, 1958).

Note that the inner max problem of (21d) is an unconstrained concave problem, and we can obtain the optimal $Q$ by taking the derivatives and setting them to 0. That is,

$$\frac{\partial L}{\partial Q_{ij}^s} = A^\pi(s, a_i) - \beta D_{ij} + \frac{\omega_j^s}{\rho_v^\pi(s)} - \frac{\beta}{\lambda}(\log Q_{ij}^s + 1) = 0, \quad \forall i, j = 1, \cdots, N, s \in \mathcal{S}. \tag{22}$$

Therefore, we have the optimal $Q_{ij}^{s*}$ as:

$$Q_{ij}^{s*} = \exp\left(\frac{\lambda}{\beta} A^\pi(s, a_i) - \lambda D_{ij}\right) \exp\left(\frac{\lambda \omega_j^s}{\beta \rho_v^\pi(s)} - 1\right), \quad \forall i, j = 1, \cdots, N, s \in \mathcal{S}. \tag{23}$$

In addition, since $\sum_{i=1}^{N} Q_{ij}^{s*} = \pi(a_j|s)$, we have the following hold:

$$\exp\left(\frac{\lambda \omega_j^s}{\beta \rho_v^\pi(s)} - 1\right) = \frac{\pi(a_j|s)}{\sum_{i=1}^{N} \exp\left(\frac{\lambda}{\beta} A^\pi(s, a_i) - \lambda D_{ij}\right)}. \tag{24}$$

By substituting the left hand side of (24) into (23), we can further reformulate the optimal $Q_{ij}^{s*}$ as:

$$Q_{ij}^{s*} = \frac{\exp\left(\frac{\lambda}{\beta} A^\pi(s, a_i) - \lambda D_{ij}\right)}{\sum_{k=1}^{N} \exp\left(\frac{\lambda}{\beta} A^\pi(s, a_k) - \lambda D_{kj}\right)} \pi(a_j|s), \quad \forall i, j = 1, \cdots, N, s \in \mathcal{S}. \tag{25}$$

To obtain the optimal dual variables, based on (24), we have the optimal $\omega^*$ as:

$$\omega_j^{s*} = \rho_v^\pi(s)\{\frac{\beta}{\lambda} + \frac{\beta}{\lambda}\ln(\pi(a_j|s)) - \frac{\beta}{\lambda}\ln[\sum_{i=1}^N \exp\left(\frac{\lambda}{\beta}A^\pi(s,a_i) - \lambda D_{ij}\right)]\}, \quad \forall j = 1, \cdots, N, s \in \mathcal{S} \qquad (26)$$

By substituting (25) and (26) into (21d), we can obtain the optimal $\beta^*$ via:

$$\min_{\beta \geq 0} \quad \beta\delta - \mathbb{E}_{s \sim \rho_v^\pi}\sum_{j=1}^N \pi(a_j|s)\{\frac{\beta}{\lambda} + \frac{\beta}{\lambda}\ln(\pi(a_j|s)) - \frac{\beta}{\lambda}\ln[\sum_{i=1}^N \exp\left(\frac{\lambda}{\beta}A^\pi(s,a_i) - \lambda D_{ij}\right)]\}$$

$$+ \mathbb{E}_{s \sim \rho_v^\pi}\sum_{i=1}^N\sum_{j=1}^N \frac{\beta}{\lambda}\frac{\exp\left(\frac{\lambda}{\beta}A^\pi(s,a_i) - \lambda D_{ij}\right) \cdot \pi(a_j|s)}{\sum_{k=1}^N \exp\left(\frac{\lambda}{\beta}A^\pi(s,a_k) - \lambda D_{kj}\right)}.$$

The proof for the upper bound of sinkhorn optimal $\beta$ can be found in Appendix E. $\qquad\square$

# E   Upper bound of Sinkhorn Optimal Beta

In this section, we will derive the upper bound of Sinkhorn optimal $\beta$. First, for a given $\beta$, the optimal $Q_{ij}^{s*}(\beta)$ to the Lagrangian dual $L(Q, \beta, \omega)$ can be expressed in (25). With this, we will present the following two lemmas:

**Lemma 1.** *The objective function (20a) with respect to $Q_{ij}^{s*}(\beta)$ decreases as the dual variable $\beta$ increases.*

**Lemma 2.** *If Assumption 1 holds, then for every $\delta > 0$, $Q_{ij}^{s*}(\frac{2A^{max}}{\delta})$ is feasible to (20b) for any $\lambda$.*

We provide proofs for Lemma 1 and Lemma 2 in Appendix E.1 and Appendix E.2 respectively. Given the above two lemmas, we are able to prove the following proposition on the upper bound of Sinkhorn optimal $\beta$:

**Proposition 2.** *If $\beta_\lambda^*$ is the optimal dual solution to the Sinkhorn dual formulation (9), then $\beta_\lambda^* \leq \frac{2A^{max}}{\delta}$ for any $\lambda$.*

*Proof of Proposition 2.* We will prove it by contradiction. According to Lemma 2, $Q_{ij}^{s*}(\frac{2A^{\max}}{\delta})$ is feasible to (20b). Since $\beta_\lambda^*$ is the optimal dual solution, $Q_{ij}^{s*}(\beta_\lambda^*)$ is optimal to (20). If $\beta_\lambda^* > \frac{2A^{\max}}{\delta}$, according to Lemma 1, the objective value in (20a) with respect to $\frac{2A^{\max}}{\delta}$ is smaller than the objective value in (20a) with respect to $\beta_\lambda^*$, which contradicts the fact that $Q_{ij}^{s*}(\beta_\lambda^*)$ is the optimal solution to (20). $\qquad\square$

## E.1   Proof of Lemma 1

**Lemma 1.** *The objective function (20a) with respect to $Q_{ij}^{s*}(\beta)$ decreases as the dual variable $\beta$ increases.*

*Proof of Lemma 1.* Let $G_\lambda(\beta)$ represent the objective function (20a). By substituting the optimal $Q_{ij}^{s*}$ in (25) into (20a), we have:

$$G_\lambda(\beta) = \mathbb{E}_{s \sim \rho_v^\pi}[\sum_{i=1}^N A^\pi(s,a_i)\sum_{j=1}^N \frac{\exp\left(\frac{\lambda}{\beta}A^\pi(s,a_i) - \lambda D_{ij}\right)}{\sum_{k=1}^N \exp\left(\frac{\lambda}{\beta}A^\pi(s,a_k) - \lambda D_{kj}\right)}\pi(a_j|s)] \qquad (27a)$$

$$= \mathbb{E}_{s \sim \rho_v^\pi}[\sum_{j=1}^N \pi(a_j|s)\sum_{i=1}^N A^\pi(s,a_i)\frac{\exp\left(\frac{\lambda}{\beta}A^\pi(s,a_i) - \lambda D_{ij}\right)}{\sum_{k=1}^N \exp\left(\frac{\lambda}{\beta}A^\pi(s,a_k) - \lambda D_{kj}\right)}]. \qquad (27b)$$

For any $\beta_2 > \beta_1 > 0$, we have:

$$G_\lambda(\beta_1) - G_\lambda(\beta_2)$$

$$= \mathbb{E}_{s \sim \rho_v^\pi}\sum_{j=1}^N \pi(a_j|s)\sum_{i=1}^N A^\pi(s,a_i)\{\frac{\exp\left(\frac{\lambda}{\beta_1}A^\pi(s,a_i) - \lambda D_{ij}\right)}{\sum_{k=1}^N \exp\left(\frac{\lambda}{\beta_1}A^\pi(s,a_k) - \lambda D_{kj}\right)}$$

$$- \frac{\exp\left(\frac{\lambda}{\beta_2} A^\pi(s, a_i) - \lambda D_{ij}\right)}{\sum_{k=1}^N \exp\left(\frac{\lambda}{\beta_2} A^\pi(s, a_k) - \lambda D_{kj}\right)}\bigg\} \tag{28a}$$

$$= \mathbb{E}_{s \sim \rho_v^\pi} \sum_{j=1}^N \pi(a_j|s) \sum_{i=1}^N A^\pi(s, a_{[i]}) \bigg\{ \frac{\exp\left(\frac{\lambda}{\beta_1} A^\pi(s, a_{[i]}) - \lambda D_{[i]j}\right)}{\sum_{k=1}^N \exp\left(\frac{\lambda}{\beta_1} A^\pi(s, a_{[k]}) - \lambda D_{[k]j}\right)}$$

$$- \frac{\exp\left(\frac{\lambda}{\beta_2} A^\pi(s, a_{[i]}) - \lambda D_{[i]j}\right)}{\sum_{k=1}^N \exp\left(\frac{\lambda}{\beta_2} A^\pi(s, a_{[k]}) - \lambda D_{[k]j}\right)}\bigg\}, \tag{28b}$$

where $[i]$ denotes sorted indices that satisfy $A^\pi(s, a_{[1]}) \geq A^\pi(s, a_{[2]}) \geq \cdots \geq A^\pi(s, a_{[N]})$. Let

$$f_s(i) = \frac{\exp\left(\frac{\lambda}{\beta_1} A^\pi(s, a_{[i]}) - \lambda D_{[i]j}\right)}{\sum_{k=1}^N \exp\left(\frac{\lambda}{\beta_1} A^\pi(s, a_{[k]}) - \lambda D_{[k]j}\right)} - \frac{\exp\left(\frac{\lambda}{\beta_2} A^\pi(s, a_{[i]}) - \lambda D_{[i]j}\right)}{\sum_{k=1}^N \exp\left(\frac{\lambda}{\beta_2} A^\pi(s, a_{[k]}) - \lambda D_{[k]j}\right)} \tag{29a}$$

$$= \frac{\exp\left(\left(\frac{\lambda}{\beta_1} - \frac{\lambda}{\beta_2}\right) A^\pi(s, a_{[i]})\right) \exp\left(\frac{\lambda}{\beta_2} A^\pi(s, a_{[i]}) - \lambda D_{[i]j}\right)}{\sum_{k=1}^N \exp\left(\left(\frac{\lambda}{\beta_1} - \frac{\lambda}{\beta_2}\right) A^\pi(s, a_{[k]})\right) \exp\left(\frac{\lambda}{\beta_2} A^\pi(s, a_{[k]}) - \lambda D_{[k]j}\right)}$$

$$- \frac{\exp\left(\frac{\lambda}{\beta_2} A^\pi(s, a_{[i]}) - \lambda D_{[i]j}\right)}{\sum_{k=1}^N \exp\left(\frac{\lambda}{\beta_2} A^\pi(s, a_{[k]}) - \lambda D_{[k]j}\right)}. \tag{29b}$$

For notation brevity, we let $m_s(i) = \exp\left(\left(\frac{\lambda}{\beta_1} - \frac{\lambda}{\beta_2}\right) A^\pi(s, a_{[i]})\right) > 0$, $w_s(i) = \exp\left(\frac{\lambda}{\beta_2} A^\pi(s, a_{[i]}) - \lambda D_{[i]j}\right) > 0$ and $q_s(i) = \frac{1}{\sum_{k=1}^N m_s(k)w_s(k)} - \frac{1}{\sum_{k=1}^N m_s(i)w_s(k)}$. Then we have

$$(29b) = \frac{m_s(i)w_s(i)}{\sum_{k=1}^N m_s(k)w_s(k)} - \frac{w_s(i)}{\sum_{k=1}^N w_s(k)} \tag{30a}$$

$$= m_s(i)w_s(i)\left(\frac{1}{\sum_{k=1}^N m_s(k)w_s(k)} - \frac{1}{\sum_{k=1}^N m_s(i)w_s(k)}\right) \tag{30b}$$

$$= m_s(i)w_s(i)q_s(i). \tag{30c}$$

Since $\frac{\lambda}{\beta_1} - \frac{\lambda}{\beta_2} > 0$, $m_s(i)$ decreases as $i$ increases. Thus, $q_s(i)$ decreases as $i$ increases. Since $m_s(1) \geq m_s(k)$ and $m_s(N) \leq m_s(k)$ for all $k = 1, \ldots, N$, we have $q_s(1) = \frac{1}{\sum_{k=1}^N m_s(k)w_s(k)} - \frac{1}{\sum_{k=1}^N m_s(1)w_s(k)} \geq \frac{1}{\sum_{k=1}^N m_s(k)w_s(k)} - \frac{1}{\sum_{k=1}^N m_s(k)w_s(k)} = 0$, and $q_s(N) = \frac{1}{\sum_{k=1}^N m_s(k)w_s(k)} - \frac{1}{\sum_{k=1}^N m_s(N)w_s(k)} \leq \frac{1}{\sum_{k=1}^N m_s(k)w_s(k)} - \frac{1}{\sum_{k=1}^N m_s(k)w_s(k)} = 0$. Since $q_s(1) \geq 0$, $q_s(N) \leq 0$ and $q_s(i)$ decreases as $i$ increases, there exists an index $1 \leq k_s \leq N$ such that $q_s(i) \geq 0$ for $i \leq k_s$ and $q_s(i) < 0$ for $i > k_s$. Since $m_s(i), w_s(i) > 0$, we have $f_s(i) \geq 0$ for $i \leq k_s$ and $f_s(i) < 0$ for $i > k_s$. In addition, we have $\sum_{i=1}^N f_s(i) = 0$ directly follows from the definition. Thus, $\sum_{i=1}^N f_s(i) = \sum_{i=1}^{k_s} |f_s(i)| - \sum_{i=k_s+1}^N |f_s(i)| = 0$. Therefore,

$$G_\lambda(\beta_1) - G_\lambda(\beta_2) = \mathbb{E}_{s \sim \rho_v^\pi} \sum_{j=1}^N \pi(a_j|s) \sum_{i=1}^N A^\pi(s, a_{[i]}) f_s(i) \tag{31a}$$

$$= \mathbb{E}_{s \sim \rho_v^\pi} \sum_{j=1}^N \pi(a_j|s) \bigg\{ \sum_{i=1}^{k_s} A^\pi(s, a_{[i]})|f_s(i)| - \sum_{i=k_s+1}^N A^\pi(s, a_{[i]})|f_s(i)| \bigg\} \tag{31b}$$

$$\geq \mathbb{E}_{s \sim \rho_v^\pi} \sum_{j=1}^N \pi(a_j|s) \bigg\{ \sum_{i=1}^{k_s} A^\pi(s, a_{[k_s]})|f_s(i)| - \sum_{i=k_s+1}^N A^\pi(s, a_{[k_s+1]})|f_s(i)| \bigg\} \tag{31c}$$

$$= \mathbb{E}_{s \sim \rho_v^\pi} \sum_{j=1}^N \pi(a_j|s) \bigg\{ A^\pi(s, a_{[k_s]}) \sum_{i=1}^{k_s} |f_s(i)| - A^\pi(s, a_{[k_s+1]}) \sum_{i=k_s+1}^N |f_s(i)| \bigg\} \tag{31d}$$

$$= \mathbb{E}_{s \sim \rho_v^\pi} \sum_{j=1}^N \pi(a_j|s) \bigg\{ A^\pi(s, a_{[k_s]}) \sum_{i=1}^{k_s} |f_s(i)| - A^\pi(s, a_{[k_s+1]}) \sum_{i=1}^{k_s} |f_s(i)| \bigg\} \tag{31e}$$

$$= \mathbb{E}_{s \sim \rho_v^\pi} \sum_{j=1}^{N} \pi(a_j|s)(A^\pi(s, a_{[k_s]}) - A^\pi(s, a_{[k_s+1]})) \sum_{i=1}^{k_s} |f_s(i)| \tag{31f}$$

$$\geq 0. \tag{31g}$$

where (31c) and (31g) hold since $A^\pi(s, a_{[i]})$ is non-increasing as $i$ increases. Furthermore, at least one inequality of (31c) and (31g) will not hold at equality since $\sum_{i=1}^{N} \pi(a_i|s)A^\pi(s, a_i) = 0$, $\forall s \in \mathcal{S}$, and for non-trivial cases, $Pr\{A^\pi(s, a) = 0, \forall s \in \mathcal{S}, \forall a \in \mathcal{A}\} < 1$, which means $Pr\{\exists s_1, s_2 \in \mathcal{S}, a_1, a_2 \in \mathcal{A}, \ s.t. \ A^\pi(s_1, a_1) \neq A^\pi(s_2, a_2)\} > 0$. Therefore, we have $G_\lambda(\beta_1) - G_\lambda(\beta_2) > 0$. □

### E.2 Proof of Lemma 2

**Lemma 2.** *If Assumption 1 holds, then for every $\delta > 0$, $Q_{ij}^{s*}(\frac{2A^{max}}{\delta})$ is feasible to (20b) for any $\lambda$.*

*Proof of Lemma 2.* By substituting the optimal $Q_{ij}^{s*}$ in (25) into (20b), we can reformulate the left hand side of (20b) as follows:

$$\mathbb{E}_{s \sim \rho_v^\pi}[\sum_{i=1}^{N} \sum_{j=1}^{N} D_{ij} Q_{ij}^{s*} + \frac{1}{\lambda} Q_{ij}^{s*} \log Q_{ij}^{s*}] \tag{32a}$$

$$= \mathbb{E}_{s \sim \rho_v^\pi}\{\sum_{i=1}^{N} \sum_{j=1}^{N} D_{ij} Q_{ij}^{s*} + \frac{1}{\lambda} Q_{ij}^{s*}[\frac{\lambda}{\beta} A^\pi(s, a_i) - \lambda D_{ij} + \log \frac{\pi(a_j|s)}{\sum_{k=1}^{N} \exp(\frac{\lambda}{\beta} A^\pi(s, a_k) - \lambda D_{kj})}]\} \tag{32b}$$

$$= \mathbb{E}_{s \sim \rho_v^\pi}\{\sum_{i=1}^{N} \sum_{j=1}^{N} \frac{1}{\beta} Q_{ij}^{s*} A^\pi(s, a_i) + \frac{1}{\lambda} Q_{ij}^{s*} \log \frac{\pi(a_j|s)}{\sum_{k=1}^{N} \exp(\frac{\lambda}{\beta} A^\pi(s, a_k) - \lambda D_{kj})}\}. \tag{32c}$$

Now we prove that when $\beta = \frac{2A^{\max}}{\delta}$, $\mathbb{E}_{s \sim \rho_v^\pi}\{\sum_{i=1}^{N} \sum_{j=1}^{N} \frac{1}{\beta} Q_{ij}^{s*}(\beta) A^\pi(s, a_i)\} \leq \frac{\delta}{2}$ and $\mathbb{E}_{s \sim \rho_v^\pi}\{\frac{1}{\lambda} Q_{ij}^{s*}(\beta) \log \frac{\pi(a_j|s)}{\sum_{k=1}^{N} \exp(\frac{\lambda}{\beta} A^\pi(s, a_k) - \lambda D_{kj})}\} \leq \frac{\delta}{2}$ hold. For the first part, we have:

$$\mathbb{E}_{s \sim \rho_v^\pi}\{\sum_{i=1}^{N} \sum_{j=1}^{N} \frac{1}{\beta} Q_{ij}^{s*} A^\pi(s, a_i)\} \tag{33a}$$

$$= \frac{1}{\beta} \mathbb{E}_{s \sim \rho_v^\pi}\{\sum_{i=1}^{N}[\sum_{j=1}^{N} Q_{ij}^{s*}] A^\pi(s, a_i)\} \tag{33b}$$

$$= \frac{1}{\beta} \mathbb{E}_{s \sim \rho_v^\pi}\{\sum_{i=1}^{N} \pi'(a_i|s) A^\pi(s, a_i)\} \tag{33c}$$

$$\leq \frac{1}{\beta} \mathbb{E}_{s \sim \rho_v^\pi}\{\sum_{i=1}^{N} \pi'(a_i|s) |A^\pi(s, a_i)|\} \tag{33d}$$

$$\leq \frac{A^{max}}{\beta} = \frac{\delta}{2}. \tag{33e}$$

For the second part, the followings hold:

$$\mathbb{E}_{s \sim \rho_v^\pi}\{\sum_{i=1}^{N} \sum_{j=1}^{N} \frac{1}{\lambda} Q_{ij}^{s*} \log \frac{\pi(a_j|s)}{\sum_{k=1}^{N} \exp(\frac{\lambda}{\beta} A^\pi(s, a_k) - \lambda D_{kj})}\} \tag{34a}$$

$$= \mathbb{E}_{s \sim \rho_v^\pi}\{\sum_{j=1}^{N} \frac{1}{\lambda}(\sum_{i=1}^{N} Q_{ij}^{s*}) \log \frac{\pi(a_j|s)}{\sum_{k=1}^{N} \exp(\frac{\lambda}{\beta} A^\pi(s, a_k) - \lambda D_{kj})}\} \tag{34b}$$

$$= \frac{1}{\lambda} \mathbb{E}_{s \sim \rho_v^\pi}\{\sum_{j=1}^{N} \pi(a_j|s) \log \frac{\pi(a_j|s)}{\sum_{k=1}^{N} \exp(\frac{\lambda}{\beta} A^\pi(s, a_k) - \lambda D_{kj})}\} \tag{34c}$$

$$\leq \frac{1}{\lambda}\mathbb{E}_{s\sim\rho_v^\pi}\{\sum_{j=1}^{N}\pi(a_j|s)\log\frac{\pi(a_j|s)}{\exp{(\frac{\lambda}{\beta}A^\pi(s,a_j))}}\} \tag{34d}$$

$$\leq \frac{1}{\lambda}\mathbb{E}_{s\sim\rho_v^\pi}\{\sum_{j=1}^{N}\pi(a_j|s)\log\frac{1}{\exp{(\frac{\lambda}{\beta}A^\pi(s,a_j))}}\} \tag{34e}$$

$$= \frac{1}{\lambda}\mathbb{E}_{s\sim\rho_v^\pi}\{\sum_{j=1}^{N}\pi(a_j|s)(-\frac{\lambda}{\beta}A^\pi(s,a_j))\} \tag{34f}$$

$$\leq \frac{1}{\beta}\mathbb{E}_{s\sim\rho_v^\pi}\{\sum_{j=1}^{N}\pi(a_j|s)|A^\pi(s,a_j)|\} \tag{34g}$$

$$\leq \frac{A^{max}}{\beta} = \frac{\delta}{2}. \tag{34h}$$

Therefore, $Q_{ij}^{s*}(\frac{2A^{\max}}{\delta})$ is feasible to (20b) for any $\lambda$. $\qquad\square$

## F   Gradient of the Objective in the Sinkhorn Dual Formulation

The closed-form gradient of the objective in the Sinkhorn dual formulation (9) is as follows:

$$\delta - \mathbb{E}_{s\sim\rho_v^\pi}\sum_{j=1}^{N}\pi(a_j|s)\Big\{\frac{1}{\lambda} + \frac{1}{\lambda}\ln(\pi(a_j|s)) - \frac{1}{\lambda}\ln[\sum_{i=1}^{N}\exp{(\frac{\lambda}{\beta}A^\pi(s,a_i)-\lambda D_{ij})]}$$

$$-\frac{\beta}{\lambda}\cdot\frac{1}{\sum_{i=1}^{N}\exp{(\frac{\lambda}{\beta}A^\pi(s,a_i)-\lambda D_{ij})}}\times\sum_{i=1}^{N}[\exp{(\frac{\lambda}{\beta}A^\pi(s,a_i)-\lambda D_{ij})}\times -\lambda A^\pi(s,a_i)\beta^{-2}]\Big\}$$

$$+\mathbb{E}_{s\sim\rho_v^\pi}\sum_{i=1}^{N}\sum_{j=1}^{N}\Big\{\frac{\pi(a_j|s)}{\lambda}\frac{\exp{(\frac{\lambda}{\beta}A^\pi(s,a_i)-\lambda D_{ij})}}{\sum_{k=1}^{N}\exp{(\frac{\lambda}{\beta}A^\pi(s,a_k)-\lambda D_{kj})}}$$

$$+\frac{\beta\pi(a_j|s)}{\lambda}\cdot\frac{\exp{(\frac{\lambda}{\beta}A^\pi(s,a_i)-\lambda D_{ij})}\times-\lambda A^\pi(s,a_i)\beta^{-2}\times\sum_{k=1}^{N}\exp{(\frac{\lambda}{\beta}A^\pi(s,a_k)-\lambda D_{kj})}}{(\sum_{k=1}^{N}\exp{(\frac{\lambda}{\beta}A^\pi(s,a_k)-\lambda D_{kj})})^2}$$

$$-\frac{\beta\pi(a_j|s)}{\lambda}\cdot\frac{\exp{(\frac{\lambda}{\beta}A^\pi(s,a_i)-\lambda D_{ij})}\times\sum_{k=1}^{N}[\exp{(\frac{\lambda}{\beta}A^\pi(s,a_k)-\lambda D_{kj})}\times-\lambda A^\pi(s,a_k)\beta^{-2}]}{(\sum_{k=1}^{N}\exp{(\frac{\lambda}{\beta}A^\pi(s,a_k)-\lambda D_{kj})})^2}\Big\}.$$

## G   Proof of Theorem 3

Given the upper bound of Wasserstein optimal $\beta$ in Theorem 1 and the upper bound of Sinkhorn optimal $\beta$ in Proposition 2, we are able to derive the following theorem:

**Theorem 3.** *Define* $\beta_{UB} = \max\{\frac{2A^{max}}{\delta},\bar{\beta}\}$. *We have:*

1. $F_\lambda(\beta)$ *converges to* $F(\beta)$ *uniformly on* $[0,\beta_{UB}]$: $\lim_{\lambda\to\infty}\sup_{0\leq\beta\leq\beta_{UB}}\left|F_\lambda(\beta)-F(\beta)\right| \leq \lim_{\lambda\to\infty}\frac{\beta_{UB}}{\lambda}N\ln N = 0$.

2. $\lim_{\lambda\to\infty} argmin_{0\leq\beta\leq\beta_{UB}}F_\lambda(\beta) \subseteq argmin_{0\leq\beta\leq\beta_{UB}}F(\beta)$.

*Proof of Theorem 3.* To show that $F_\lambda(\beta)$ converges to $F(\beta)$ uniformly on $[0,\beta_{\mathrm{UB}}]$, it is equivalent to show that $\lim_{\lambda\to\infty}\sup_{0\leq\beta\leq\beta_{\mathrm{UB}}}\left|F_\lambda(\beta)-F(\beta)\right| = 0$. Let $\epsilon_s^\pi(\beta,i,j) = \max_{k=1...N}(A^\pi(s,a_k)-\beta D_{kj})-[A^\pi(s,a_i)-\beta D_{ij}]$, and $\epsilon_s^\pi(\beta,i,j)\geq 0$. First, we have

$$\left|F_\lambda(\beta)-F(\beta)\right|$$

$$= \left| \beta\delta - \mathbb{E}_{s\sim\rho_v^\pi} \sum_{j=1}^N \pi(a_j|s)\{\frac{\beta}{\lambda} + \frac{\beta}{\lambda}\ln(\pi(a_j|s)) - \frac{\beta}{\lambda}\ln[\sum_{i=1}^N \exp{(\frac{\lambda}{\beta}A^\pi(s,a_i) - \lambda D_{ij})}]\} \right.$$

$$+ \mathbb{E}_{s\sim\rho_v^\pi} \sum_{i=1}^N \sum_{j=1}^N \frac{\beta}{\lambda} \frac{\exp{(\frac{\lambda}{\beta}A^\pi(s,a_i) - \lambda D_{ij})} \cdot \pi(a_j|s)}{\sum_{k=1}^N \exp{(\frac{\lambda}{\beta}A^\pi(s,a_k) - \lambda D_{kj})}} - \beta\delta$$

$$\left. - \mathbb{E}_{s\sim\rho_v^\pi} \sum_{j=1}^N \pi(a_j|s)\max_{i=1...N}(A^\pi(s,a_i) - \beta D_{ij}) \right| \tag{35a}$$

$$\leq \left| \frac{\beta}{\lambda}\mathbb{E}_{s\sim\rho_v^\pi} \sum_{j=1}^N \pi(a_j|s) \right| + \left| \frac{\beta}{\lambda}\mathbb{E}_{s\sim\rho_v^\pi} \sum_{j=1}^N \pi(a_j|s)\ln(\pi(a_j|s)) \right|$$

$$+ \left| \mathbb{E}_{s\sim\rho_v^\pi} \sum_{i=1}^N \sum_{j=1}^N \frac{\beta}{\lambda} \frac{\exp{(\frac{\lambda}{\beta}A^\pi(s,a_i) - \lambda D_{ij})} \cdot \pi(a_j|s)}{\sum_{k=1}^N \exp{(\frac{\lambda}{\beta}A^\pi(s,a_k) - \lambda D_{kj})}} \right|$$

$$+ \left| \mathbb{E}_{s\sim\rho_v^\pi} \sum_{j=1}^N \pi(a_j|s)\frac{\beta}{\lambda}\ln[\sum_{i=1}^N \exp{(\frac{\lambda}{\beta}A^\pi(s,a_i) - \lambda D_{ij})}] \right.$$

$$\left. - \mathbb{E}_{s\sim\rho_v^\pi} \sum_{j=1}^N \pi(a_j|s)\max_{i=1...N}(A^\pi(s,a_i) - \beta D_{ij}) \right| \tag{35b}$$

$$\leq 2\left| \frac{\beta}{\lambda}\mathbb{E}_{s\sim\rho_v^\pi} \sum_{j=1}^N \pi(a_j|s) \right| + \left| \frac{\beta}{\lambda}\mathbb{E}_{s\sim\rho_v^\pi} \sum_{j=1}^N \pi(a_j|s)\ln(\pi(a_j|s)) \right|$$

$$+ \left| \mathbb{E}_{s\sim\rho_v^\pi} \sum_{j=1}^N \pi(a_j|s)\frac{\beta}{\lambda}\ln[\sum_{i=1}^N \exp{(\frac{\lambda}{\beta}A^\pi(s,a_i) - \lambda D_{ij})}] \right.$$

$$\left. - \mathbb{E}_{s\sim\rho_v^\pi} \sum_{j=1}^N \pi(a_j|s)\max_{i=1...N}(A^\pi(s,a_i) - \beta D_{ij}) \right|. \tag{35c}$$

In addition,

$$\left| \mathbb{E}_{s\sim\rho_v^\pi} \sum_{j=1}^N \pi(a_j|s)\frac{\beta}{\lambda}\ln[\sum_{i=1}^N \exp{(\frac{\lambda}{\beta}A^\pi(s,a_i) - \lambda D_{ij})}] \right.$$

$$\left. - \mathbb{E}_{s\sim\rho_v^\pi} \sum_{j=1}^N \pi(a_j|s)\max_{i=1...N}(A^\pi(s,a_i) - \beta D_{ij}) \right| \tag{36a}$$

$$= \left| \mathbb{E}_{s\sim\rho_v^\pi} \sum_{j=1}^N \pi(a_j|s)\frac{\beta}{\lambda}\ln[\exp{(\frac{\lambda}{\beta}\max_{k=1...N}(A^\pi(s,a_k) - \beta D_{kj}))}\sum_{i=1}^N \exp{(-\frac{\lambda}{\beta}\epsilon_s^\pi(\beta,i,j))}] \right.$$

$$\left. - \mathbb{E}_{s\sim\rho_v^\pi} \sum_{j=1}^N \pi(a_j|s)\max_{i=1...N}(A^\pi(s,a_i) - \beta D_{ij}) \right| \tag{36b}$$

$$= \left| \mathbb{E}_{s\sim\rho_v^\pi} \sum_{j=1}^N \pi(a_j|s)\frac{\beta}{\lambda}\{\ln[\exp{(\frac{\lambda}{\beta}\max_{k=1...N}(A^\pi(s,a_k) - \beta D_{kj}))}] + \ln[\sum_{i=1}^N \exp{(-\frac{\lambda}{\beta}\epsilon_s^\pi(\beta,i,j))}]\} \right.$$

$$\left. - \mathbb{E}_{s\sim\rho_v^\pi} \sum_{j=1}^N \pi(a_j|s)\max_{i=1...N}(A^\pi(s,a_i) - \beta D_{ij}) \right| \tag{36c}$$

$$= \left| \mathbb{E}_{s\sim\rho_v^\pi} \sum_{j=1}^N \pi(a_j|s)\frac{\beta}{\lambda}\ln[\sum_{i=1}^N \exp{(-\frac{\lambda}{\beta}\epsilon_s^\pi(\beta,i,j))}] \right|. \tag{36d}$$

Therefore,

$$\lim_{\lambda \to \infty} \sup_{0 \leq \beta \leq \beta_{\text{UB}}} \left| F_\lambda(\beta) - F(\beta) \right| \tag{37a}$$

$$\leq \lim_{\lambda \to \infty} \frac{2\beta_{\text{UB}}}{\lambda} \left| \mathbb{E}_{s \sim \rho_v^\pi} \sum_{j=1}^N \pi(a_j|s) \right| + \lim_{\lambda \to \infty} \frac{\beta_{\text{UB}}}{\lambda} \left| \mathbb{E}_{s \sim \rho_v^\pi} \sum_{j=1}^N \pi(a_j|s) \ln(\pi(a_j|s)) \right|$$

$$+ \lim_{\lambda \to \infty} \sup_{0 \leq \beta \leq \beta_{\text{UB}}} \frac{\beta}{\lambda} \left| \mathbb{E}_{s \sim \rho_v^\pi} \sum_{j=1}^N \pi(a_j|s) \ln[\sum_{i=1}^N \exp{(-\frac{\lambda}{\beta} \epsilon_s^\pi(\beta, i, j))}] \right| \tag{37b}$$

$$= \lim_{\lambda \to \infty} \sup_{0 \leq \beta \leq \beta_{\text{UB}}} \frac{\beta}{\lambda} \left| \mathbb{E}_{s \sim \rho_v^\pi} \sum_{j=1}^N \pi(a_j|s) \ln[\sum_{i=1}^N \exp{(-\frac{\lambda}{\beta} \epsilon_s^\pi(\beta, i, j))}] \right|. \tag{37c}$$

In addition, $\forall \beta \in [0, \beta_{\text{UB}}]$ and $\forall \lambda$, $\epsilon_s^\pi(\beta, i, j)$ is bounded since

$$\left| \epsilon_s^\pi(\beta, i, j) \right| = \left| \max_{k=1...N} (A^\pi(s, a_k) - \beta D_{kj}) - [A^\pi(s, a_i) - \beta D_{ij}] \right| \tag{38}$$

$$\leq 2 \max_{s,a} A^\pi(s, a) + \beta_{\text{UB}} \max_{i,j} D_{ij}$$

$$\leq 2A^{\max} + \beta_{\text{UB}} \max_{i,j} D_{ij} < \infty. \tag{39}$$

Then, $\left| \mathbb{E}_{s \sim \rho_v^\pi} \sum_{j=1}^N \pi(a_j|s) \ln[\sum_{i=1}^N \exp{(-\frac{\lambda}{\beta} \epsilon_s^\pi(\beta, i, j))}] \right|$ is bounded. Therefore in (37c), the optimal $\beta$ can be achieved. Let $\beta^\lambda = \text{argmax}_{0 \leq \beta \leq \beta_{\text{UB}}} \frac{\beta}{\lambda} \left| \mathbb{E}_{s \sim \rho_v^\pi} \sum_{j=1}^N \pi(a_j|s) \ln[\sum_{i=1}^N \exp{(-\frac{\lambda}{\beta} \epsilon_s^\pi(\beta, i, j))}] \right|$, and then we have:

$$\lim_{\lambda \to \infty} \sup_{0 \leq \beta \leq \beta_{\text{UB}}} \frac{\beta}{\lambda} \left| \mathbb{E}_{s \sim \rho_v^\pi} \sum_{j=1}^N \pi(a_j|s) \ln[\sum_{i=1}^N \exp{(-\frac{\lambda}{\beta} \epsilon_s^\pi(\beta, i, j))}] \right| \tag{40a}$$

$$= \lim_{\lambda \to \infty} \frac{\beta^\lambda}{\lambda} \left| \mathbb{E}_{s \sim \rho_v^\pi} \sum_{j=1}^N \pi(a_j|s) \ln[\sum_{i=1}^N \exp{(-\frac{\lambda}{\beta^\lambda} \epsilon_s^\pi(\beta^\lambda, i, j))}] \right|. \tag{40b}$$

Let $\mathcal{K}_s^\pi(\beta, j) = \text{argmax}_{k=1...N} A^\pi(s, a_k) - \beta D_{kj}$. Define $\sigma_s(j) = \min_{0 \leq \beta \leq \beta_{\text{UB}}} \min_{i=1...N, i \notin \mathcal{K}_s^\pi(\beta, j)} \epsilon_s^\pi(\beta, i, j)$. Then since $\epsilon_s^\pi(\beta, i, j) > 0$ for $i \notin \mathcal{K}_s^\pi(\beta, j)$ based on its definition, we have $\sigma_s(j) > 0$. On one hand, we have

$$\lim_{\lambda \to \infty} \ln[\sum_{i=1}^N \exp{(-\frac{\lambda}{\beta^\lambda} \epsilon_s^\pi(\beta^\lambda, i, j))}] \tag{41a}$$

$$= \lim_{\lambda \to \infty} \ln[\sum_{i=1|i \notin \mathcal{K}_s^\pi(\beta_\lambda, j)}^N \exp{(-\frac{\lambda}{\beta^\lambda} \epsilon_s^\pi(\beta^\lambda, i, j))} + \sum_{i=1|i \in \mathcal{K}_s^\pi(\beta_\lambda, j)}^N \exp{(-\frac{\lambda}{\beta^\lambda} \epsilon_s^\pi(\beta^\lambda, i, j))}] \tag{41b}$$

$$\leq \lim_{\lambda \to \infty} \ln[\sum_{i=1|i \notin \mathcal{K}_s^\pi(\beta_\lambda, j)}^N \exp{(-\frac{\lambda}{\beta_{\text{UB}}} \sigma_s(j))} + \sum_{i=1|i \in \mathcal{K}_s^\pi(\beta_\lambda, j)}^N \exp{(0)}] \tag{41c}$$

$$= \lim_{\lambda \to \infty} \ln[\sum_{i=1|i \notin \mathcal{K}_s^\pi(\beta_\lambda, j)}^N \exp{(-\frac{\lambda}{\beta_{\text{UB}}} \sigma_s(j))} + |\mathcal{K}_s^\pi(\beta_\lambda, j)|] \tag{41d}$$

$$= \lim_{\lambda \to \infty} \ln[|\mathcal{K}_s^\pi(\beta_\lambda, j)|]. \tag{41e}$$

On the other hand, we have

$$\lim_{\lambda \to \infty} \ln[\sum_{i=1}^N \exp{(-\frac{\lambda}{\beta^\lambda} \epsilon_s^\pi(\beta^\lambda, i, j))}] \tag{42a}$$

$$= \lim_{\lambda \to \infty} \ln\Big[ \sum_{i=1 | i \notin \mathcal{K}_s^\pi(\beta_\lambda, j)}^N \exp\big(-\frac{\lambda}{\beta^\lambda} \epsilon_s^\pi(\beta^\lambda, i, j)\big) + \sum_{i=1 | i \in \mathcal{K}_s^\pi(\beta_\lambda, j)}^N \exp\big(-\frac{\lambda}{\beta^\lambda} \epsilon_s^\pi(\beta^\lambda, i, j)\big)\Big] \tag{42b}$$

$$\geq \lim_{\lambda \to \infty} \ln\Big[ \sum_{i=1 | i \in \mathcal{K}_s^\pi(\beta_\lambda, j)}^N \exp\big(-\frac{\lambda}{\beta^\lambda} \epsilon_s^\pi(\beta^\lambda, i, j)\big)\Big] \tag{42c}$$

$$= \lim_{\lambda \to \infty} \ln\Big[ \sum_{i=1 | i \in \mathcal{K}_s^\pi(\beta_\lambda, j)}^N \exp(0)\Big] \tag{42d}$$

$$= \lim_{\lambda \to \infty} \ln[|\mathcal{K}_s^\pi(\beta_\lambda, j)|]. \tag{42e}$$

Therefore, $\lim_{\lambda \to \infty} \Big| \ln[\sum_{i=1}^N \exp\big(-\frac{\lambda}{\beta^\lambda} \epsilon_s^\pi(\beta^\lambda, i, j)\big)]\Big| = \lim_{\lambda \to \infty} \ln[|\mathcal{K}_s^\pi(\beta_\lambda, j)|]$. Based on that, we have

$$\lim_{\lambda \to \infty} \frac{\beta^\lambda}{\lambda} \Big| \mathbb{E}_{s \sim \rho_v^\pi} \sum_{j=1}^N \pi(a_j|s) \ln[\sum_{i=1}^N \exp\big(-\frac{\lambda}{\beta^\lambda} \epsilon_s^\pi(\beta^\lambda, i, j)\big)]\Big| \tag{43a}$$

$$\leq \lim_{\lambda \to \infty} \frac{\beta^\lambda}{\lambda} \Big| \sum_{j=1}^N \ln[\sum_{i=1}^N \exp\big(-\frac{\lambda}{\beta^\lambda} \epsilon_s^\pi(\beta^\lambda, i, j)\big)]\Big| \tag{43b}$$

$$\leq \lim_{\lambda \to \infty} \frac{\beta^\lambda}{\lambda} \sum_{j=1}^N \Big| \ln[\sum_{i=1}^N \exp\big(-\frac{\lambda}{\beta^\lambda} \epsilon_s^\pi(\beta^\lambda, i, j)\big)]\Big| \tag{43c}$$

$$= \lim_{\lambda \to \infty} \frac{\beta^\lambda}{\lambda} \sum_{j=1}^N \ln[|\mathcal{K}_s^\pi(\beta_\lambda, j)|] \tag{43d}$$

$$\leq \lim_{\lambda \to \infty} \frac{\beta_{\mathrm{UB}}}{\lambda} N \ln N = 0, \tag{43e}$$

which means $\lim_{\lambda \to \infty} \sup_{0 \leq \beta \leq \beta_{\mathrm{UB}}} \Big| F_\lambda(\beta) - F(\beta)\Big| \leq 0$. Furthermore, since $\lim_{\lambda \to \infty} \sup_{0 \leq \beta \leq \beta_{\mathrm{UB}}} |F_\lambda(\beta) - F(\beta)| \geq 0$ holds naturally, we have $\lim_{\lambda \to \infty} \sup_{0 \leq \beta \leq \beta_{\mathrm{UB}}} |F_\lambda(\beta) - F(\beta)| = 0$. Therefore, $F_\lambda(\beta)$ converges to $F(\beta)$ uniformly on $[0, \beta_{\mathrm{UB}}]$, which also indicates $F_\lambda(\beta)$ epi-converges to $F(\beta)$ on $[0, \beta_{\mathrm{UB}}]$ (Royset, 2018; Rockafellar & Wets, 1998). By properties of epi-convergence, we have that $\lim_{\lambda \to \infty} \mathrm{argmin}_{0 \leq \beta \leq \beta_{\mathrm{UB}}} F_\lambda(\beta) \subseteq \mathrm{argmin}_{0 \leq \beta \leq \beta_{\mathrm{UB}}} F(\beta)$ (Rockafellar & Wets, 1998). $\qquad\square$

## H  Proof of Lemma 1

**Lemma 1.** *As $\lambda_k \to \infty$, SPO update converges to WPO update:* $\lim_{\lambda_k \to \infty} \mathbb{F}^{SPO}(\pi_k) \in \mathbb{F}^{WPO}(\pi_k)$.

*Proof of Lemma 1.* Let $\xi_s^k(i, j) = \frac{\lambda}{\beta_k} \{\max_{l=1,\ldots,N}(\hat{A}^{\pi_k}(s, a_l) - \beta_k D_{lj}) - [\hat{A}^{\pi_k}(s, a_i) - \beta_k D_{ij}]\}$. The SPO update with $\lambda \to \infty$ equals to:

$$\pi_{k+1}(a_i|s) = \lim_{\lambda \to \infty} \mathbb{F}^{\mathrm{SPO}}(\pi_k) = \lim_{\lambda \to \infty} \sum_{j=1}^N \frac{\exp\big(\frac{\lambda}{\beta_k} \hat{A}^{\pi_k}(s, a_i) - \lambda D_{ij}\big)}{\sum_{l=1}^N \exp\big(\frac{\lambda}{\beta_k} \hat{A}^{\pi_k}(s, a_l) - \lambda D_{lj}\big)} \pi_k(a_j|s) \tag{44a}$$

$$= \lim_{\lambda \to \infty} \sum_{j=1}^N \frac{\exp\big(\hat{A}^{\pi_k}(s, a_{\hat{k}_s^{\pi_k}(\beta_k, j)}) - \beta_k D_{\hat{k}_s^{\pi_k}(\beta_k, j)j}\big) \cdot \exp(-\xi_s^k(i, j))}{\exp\big(\hat{A}^{\pi_t}(s, a_{\hat{k}_s^{\pi_k}(\beta_k, j)}) - \beta_k D_{\hat{k}_s^{\pi_k}(\beta_k, j)j}\big) \cdot \sum_{l=1}^N \exp(-\xi_s^k(l, j))} \pi_k(a_j|s) \tag{44b}$$

$$= \lim_{\lambda \to \infty} \sum_{j=1}^N \frac{\exp\big(-\xi_s^k(i, j)\big)}{\sum_{l=1}^N \exp(-\xi_s^k(l, j))} \pi_k(a_j|s) \tag{44c}$$

$$= \lim_{\lambda \to \infty} \sum_{j=1}^N \frac{\exp\big(-\xi_s^k(i, j)\big) \cdot \pi_k(a_j|s)}{\sum_{l \in \hat{\mathcal{K}}_s^{\pi_k}(\beta_k, j)} \exp(-\xi_s^k(l, j)) + \sum_{l \notin \hat{\mathcal{K}}_s^{\pi_k}(\beta_k, j)} \exp(-\xi_s^k(l, j))} \tag{44d}$$

$$= \sum_{j=1}^{N} \frac{\lim_{\lambda \to \infty} \exp\left(-\xi_s^k(i,j)\right) \cdot \pi_k(a_j|s)}{\sum_{l \in \hat{\mathcal{K}}_s^{\pi_k}(\beta_k, j)} \lim_{\lambda \to \infty} \exp(-\xi_s^k(l,j)) + \sum_{l \notin \hat{\mathcal{K}}_s^{\pi_k}(\beta_k, j)} \lim_{\lambda \to \infty} \exp(-\xi_s^k(l,j))} \tag{44e}$$

$$= \sum_{j=1}^{N} \frac{I_{\hat{\mathcal{K}}_s^{\pi_k}(\beta_k, j)}(i)}{|\hat{\mathcal{K}}_s^{\pi_k}(\beta_k, j)|} \pi_k(a_j|s), \tag{44f}$$

where $I$ denotes the indicator function; (44f) holds because as $\lambda \to \infty$, $\xi_s^k(i,j) = \infty$ for $i \notin \hat{\mathcal{K}}_s^{\pi_k}(\beta_k, j)$ and 0 otherwise, thus $\lim_{\lambda \to \infty} \exp\left(-\xi_s^k(i,j)\right) = 0$ for $i \notin \hat{\mathcal{K}}_s^{\pi_k}(\beta_k, j)$ and 1 otherwise.

Let $f_s^k(i,j) = \frac{1}{|\hat{\mathcal{K}}_s^{\pi_k}(\beta_k, j)|}$ if $i \in \hat{\mathcal{K}}_s^{\pi_k}(\beta_k, j)$, and $f_s^k(i,j) = 0$ otherwise. Therefore, SPO update with $\lambda \to \infty$ equals to the following WPO update, $\mathbb{F}^{\mathrm{WPO}}(\pi_k) = \sum_{j=1}^{N} \pi_k(a_j|s) f_s^k(i,j)$. $\qquad \square$

## I   Proof of Theorem 4

**Theorem 4. (Performance improvement)** *For any initial state distribution $\upsilon$ and any $\beta_k \geq 0$, if $||\hat{A}^\pi - A^\pi||_\infty \leq \epsilon$ for some $\epsilon > 0$, let $\hat{\mathcal{K}}_s^{\pi_k}(\beta_k, j) = argmax_{i=1,\dots,N}\{\hat{A}^{\pi_k}(s, a_i) - \beta_k D_{ij}\}$, WPO policy update (and SPO with $\lambda \to \infty$) guarantee the following performance improvement bound when the inaccurate advantage function $\hat{A}^\pi$ is used,*

$$J(\pi_{k+1}) \geq J(\pi_k) + \beta_k \mathbb{E}_{s \sim \rho_\upsilon^{\pi_{k+1}}} \sum_{j=1}^{N} \pi_k(a_j|s) \sum_{i \in \hat{\mathcal{K}}_s^{\pi_k}(\beta_k, j)} f_s^k(i,j) D_{ij} - \frac{2\epsilon}{1-\gamma}. \tag{10}$$

*Proof of Theorem 4.*

$$J(\pi_{k+1}) - J(\pi_k) = \mathbb{E}_{s \sim \rho_\upsilon^{\pi_{k+1}}} \mathbb{E}_{a \sim \pi_{k+1}} [A^{\pi_k}(s, a)] \tag{45a}$$

$$= \mathbb{E}_{s \sim \rho_\upsilon^{\pi_{k+1}}} \sum_{i=1}^{N} \pi_{k+1}(a_i|s) A^{\pi_k}(s, a_i) \tag{45b}$$

$$= \mathbb{E}_{s \sim \rho_\upsilon^{\pi_{k+1}}} \sum_{i=1}^{N} \sum_{j=1}^{N} \pi_k(a_j|s) f_s^k(i,j) A^{\pi_k}(s, a_i) \tag{45c}$$

$$= \mathbb{E}_{s \sim \rho_\upsilon^{\pi_{k+1}}} \sum_{j=1}^{N} \pi_k(a_j|s) \sum_{i=1}^{N} f_s^k(i,j) A^{\pi_k}(s, a_i) \tag{45d}$$

$$= \mathbb{E}_{s \sim \rho_\upsilon^{\pi_{k+1}}} \sum_{j=1}^{N} \pi_k(a_j|s) \sum_{i \in \hat{\mathcal{K}}_s^{\pi_k}(\beta_k, j)} f_s^k(i,j) A^{\pi_k}(s, a_i) \tag{45e}$$

$$\geq \mathbb{E}_{s \sim \rho_\upsilon^{\pi_{k+1}}} \sum_{j=1}^{N} \pi_k(a_j|s) \sum_{i \in \hat{\mathcal{K}}_s^{\pi_k}(\beta_k, j)} f_s^k(i,j) [A^{\pi_k}(s, a_j) + \beta_k D_{ij} - 2\epsilon] \tag{45f}$$

$$= \beta_k \mathbb{E}_{s \sim \rho_\upsilon^{\pi_{k+1}}} \sum_{j=1}^{N} \pi_k(a_j|s) \sum_{i \in \hat{\mathcal{K}}_s^{\pi_k}(\beta_k, j)} f_s^k(i,j) D_{ij} - \frac{2\epsilon}{1-\gamma}, \tag{45g}$$

where (45a) holds due to the performance difference lemma in Kakade & Langford (2002); (45f) follows from the definition of $\hat{\mathcal{K}}_s^{\pi_k}(\beta_k, j)$ and the fact that $||\hat{A}^{\pi_k} - A^{\pi_k}||_\infty \leq \epsilon$, therefore for $i \in \hat{\mathcal{K}}_s^{\pi_k}(\beta_k, j)$, $[A^{\pi_k}(s, a_i) + \epsilon] - \beta_k D_{ij} \geq \hat{A}^{\pi_k}(s, a_i) - \beta_k D_{ij} \geq \hat{A}^{\pi_k}(s, a_j) - \beta_k D_{jj} = \hat{A}^{\pi_k}(s, a_j) \geq A^{\pi_k}(s, a_j) - \epsilon$; (45g) holds since $\mathbb{E}_{a \sim \pi}[A^\pi(s, a)] = 0$. $\qquad \square$

## J Proof of Theorem 5

**Theorem 5.** *(Global convergence)* *Under Assumption 2, we have for any $\beta_k \geq 0$, (WPO) satisfies that*

$$\|V^\star - V^{\pi_{k+1}}\|_\infty \leq \gamma\|V^\star - V^{\pi_k}\|_\infty + \beta_k\|D\|_\infty, \tag{11}$$

*and (SPO) satisfies that*

$$\|V^\star - V^{\pi_{k+1}}\|_\infty \leq \gamma\|V^\star - V^{\pi_k}\|_\infty + 2\frac{\beta_k}{1-\gamma}\left(\|D\|_\infty + 2\frac{\log N}{\lambda}\right). \tag{12}$$

*If $\lim_{k\to\infty} \beta_k = 0$, we further have $\lim_{k\to\infty} J(\pi_k) = J^\star$.*

*Proof of Theorem 5.* Our proof is inspired by the work Bhandari & Russo (2021).

We use the shorthand $\pi_s$ for the probability distribution $\pi(\cdot \mid s)$ on the actions and denote the probability distribution on the action space $\mathcal{A}$ as $\Delta$. To save notations, we rewrite $\pi_{k+1}, \pi_k$ and $\beta_k$ as $\pi^+, \pi$ and $\beta$ respectively. We use $d$ for either $d_W$ or $d_S$ in the following derivation. Note $d \leq \|D\|_\infty =: D$ for both cases[4], and $d_S \geq -2\frac{\log N}{\lambda}$. [5]

Since a policy $\pi$ is indeed just a member of $\prod_{i=1}^{|S|} \Delta$, we find that the problem (7) can be split into $|\mathcal{S}|$ many optimization problems. For each $s \in \mathcal{S}$, we need to solve

$$\max_{\pi'_s \in \Delta} \quad \rho^\pi(s)\mathbb{E}_{a\sim\pi'(\cdot|s)}[A^\pi(s,a)] - \beta\rho^\pi(s)d(\pi'_s, \pi_s). \tag{46}$$

Denote the quality function of $\pi$ as $Q^\pi(s,a) = \mathbb{E}[R_t|s_t = s, a_t = a; \pi]$, and the value function of $\pi$ as $V^\pi(s) = \mathbb{E}[R_t|s_t = s; \pi]$, we find that $A^\pi(s,a) = Q^\pi(s,a) - V^\pi(s)$. Since the second term is only a function of the current policy $\pi$ and the state $s$, we find that Problem (46) is further equivalent to (in the sense of the same solution set):

$$\max_{\pi'_s \in \Delta} \quad \mathbb{E}_{a\sim\pi'_s}[Q^\pi(s,a)] - \beta d(\pi'_s, \pi_s). \tag{47}$$

Here we use $\rho_0(s) > 0$ for all $s$. Let $\bar{\pi}$ be a solution of the policy iteration:

$$\bar{\pi}_s \in \arg\max_{\pi'_s} \mathbb{E}_{a\sim\pi'_s}[Q^\pi(s,a)]. \tag{48}$$

Also define the bellman operator $T : \mathbb{R}^{|\mathcal{S}|} \to \mathbb{R}^{|\mathcal{S}|}$ and the operator $T^{\pi'} : \mathbb{R}^{|\mathcal{S}|} \to \mathbb{R}^{|\mathcal{S}|}$: for any $V \in \mathbb{R}^{|\mathcal{S}|}$,

$$(TV)_s = \max_{a\in\mathcal{A}} r(s,a) + \gamma\mathbb{E}_{s'\sim P(\cdot|s,a)}[V(s')], \tag{49}$$

$$(T^{\pi'}V)_s = \mathbb{E}_{a\sim\pi'_s}[r(s,a) + \gamma\mathbb{E}_{s'\sim P(\cdot|s,a)}V(s')]. \tag{50}$$

Using the relation between the quality function and the value function, $Q^\pi(s,a) = r(s,a) + \mathbb{E}_{s'\sim P(\cdot|s,a)}[V^\pi(s')]$, we can rewrite the above equations in terms of the quality function for $V = V^\pi$:

$$(TV^\pi)_s = \max_{a\in\mathcal{A}} r(s,a) + \gamma\mathbb{E}_{s'\sim P(\cdot|s,a)}[V^\pi(s')] = \max_{a\in\mathcal{A}} Q(s,a) = T^{\bar{\pi}}V^\pi, \tag{51}$$

$$(T^{\pi'}V^\pi)_s = \mathbb{E}_{a\sim\pi'_s}[Q^\pi(s,a)]. \tag{52}$$

---

[4]For Sinkhorn divergence, note that the entropy function is always nonnegative.

[5]This lower bound is obtained via $d_S(\pi', \pi|\lambda) \geq \min_{Q\geq 0, \sum_{i,j} Q_{ij}=1}\left\{\langle Q, D\rangle - \frac{1}{\lambda}h(Q)\right\} \overset{(a)}{=} \langle Q, D\rangle - \frac{1}{\lambda}h(Q)|_{Q_{ij}=\frac{\exp(-\lambda D_{ij})}{\sum_{i,j}\exp(-\lambda D_{ij})}} = -\frac{1}{\lambda}\log\left(\sum_{i,j}\exp(-\lambda D_{ij})\right) \overset{(b)}{\geq} -\frac{2\log N}{\lambda}$. Here in the step $(a)$, we use the Lagrangian multiplier method to derive the optimal $Q_{ij} = \frac{\exp(-\lambda D_{ij})}{\sum_{i,j}\exp(-\lambda D_{ij})}$. In the step $(b)$, we use the fact that $\log(\sum_{i=1}^n \exp(x_i)) \leq \max\{x_1, \ldots, x_n\} + \log n$ for any $x_1, \ldots, x_n \in \mathbb{R}$ and $D_{ii} = 0$ for any $i$.

Let us consider $d = d_{\mathrm{W}}$ first. Using the optimality of $\pi^+$ for the problem (46), we know that

$$
\begin{aligned}
&\mathbb{E}_{a \in \pi_s^+}[Q^\pi(s,a)] - \beta d(\pi_s^+, \pi_s) \geq \mathbb{E}_{a \in \bar{\pi}_s}[Q^\pi(s,a)] - \beta d(\bar{\pi}_s, \pi_s) \\
\Longrightarrow\ &\mathbb{E}_{a \in \pi_s^+}[Q^\pi(s,a)] \geq \mathbb{E}_{a \in \bar{\pi}_s}[Q^\pi(s,a)] - \beta D.
\end{aligned}
\tag{53}
$$

and

$$
\begin{aligned}
&\mathbb{E}_{a \in \pi_s^+}[Q^\pi(s,a)] - \beta d(\pi_s^+, \pi_s) \geq \mathbb{E}_{a \in \pi_s}[Q^\pi(s,a)] - \beta d(\pi_s, \pi_s) \\
\Longrightarrow\ &\mathbb{E}_{a \in \pi_s^+}[Q^\pi(s,a)] \geq \mathbb{E}_{a \in \pi_s}[Q^\pi(s,a)] = V^\pi(s).
\end{aligned}
\tag{54}
$$

Using the notation in (51) and (52), (53) and (54) become

$$
T^{\pi^+} V^\pi \geq T V^\pi - \beta D \mathbf{1}_{|\mathcal{S}|},
\tag{55}
$$

$$
T^{\pi^+} V^\pi \geq V^\pi.
\tag{56}
$$

Here $\mathbf{1}_{|\mathcal{S}|}$ is a vector of all one entries and the inequality $\geq$ means entrywisely larger than or equal to. By iteratively applying $T^{\pi^+}$ to (56) and use the fact that $T^{\pi^+}$ is a monotone and contraction map with $V^{\pi^+}$ as the unique fixed point, we have

$$
V^{\pi^+} \geq \cdots \geq (T^{\pi^+})^2 V^\pi \geq T^{\pi^+} V^\pi \geq V^\pi.
\tag{57}
$$

Hence we have

$$
0 \overset{(a)}{\leq} V^\star - V^{\pi^+} \overset{(b)}{\leq} V^\star - T^{\pi^+} V^\pi \overset{(c)}{\leq} V^\star - T V^\pi + \beta D \mathbf{1}_{|\mathcal{S}|}.
\tag{58}
$$

Here the inequality $(a)$ is due to the optimality of $V^\star$. The inequality $(b)$ is due to (57), and the inequality $(c)$ is due to (55). Now using the fact $V^\star$ is the unique fixed point of $T$, and $T$ is a monotone and contraction map, we have from (58) that

$$
\|V^\star - V^{\pi^+}\|_\infty \leq \|T V^\star - T V^\pi\|_\infty + \beta D \leq \gamma \|V^\star - V^\pi\|_\infty + \beta D.
\tag{59}
$$

Next consider $d = d_{\mathrm{S}}$. The optimality of $\pi^+$ reveals that for $\tilde{\pi} = \bar{\pi}$ or $\pi$:

$$
\begin{aligned}
&\mathbb{E}_{a \in \pi_s^+}[Q^\pi(s,a)] - \beta d(\pi_s^+, \pi_s) \geq \mathbb{E}_{a \in \tilde{\pi}_s}[Q^\pi(s,a)] - \beta d(\tilde{\pi}_s, \pi_s) \\
\Longrightarrow\ &\mathbb{E}_{a \in \pi_s^+}[Q^\pi(s,a)] \geq \mathbb{E}_{a \in \tilde{\pi}_s}[Q^\pi(s,a)] - \beta(D + 2\frac{\log N}{\lambda}).
\end{aligned}
\tag{60}
$$

Thus we have the following

$$
T^{\pi^+} V^\pi \geq T V^\pi - \beta(D + \frac{2 \log N}{\lambda}) \mathbf{1}_{|\mathcal{S}|},
\tag{61}
$$

$$
T^{\pi^+} V^\pi \geq V^\pi - \beta(D + \frac{2 \log N}{\lambda}) \mathbf{1}_{|\mathcal{S}|}.
\tag{62}
$$

By iteratively applying $T^{\pi^+}$ to (62) and use the fact that $T^{\pi^+}$ is a monotone and contraction map with $V^{\pi^+}$ as the unique fixed point, we have

$$
V^{\pi^+} \geq V^\pi - \frac{\beta}{1-\gamma}(D + 2\frac{\log N}{\lambda}) \mathbf{1}_{|\mathcal{S}|}.
\tag{63}
$$

Hence we have

$$
\begin{aligned}
0 &\overset{(a)}{\leq} V^\star - V^{\pi^+} \overset{(b)}{\leq} V^\star - T^{\pi^+} V^\pi + \frac{\beta}{1-\gamma}(D + 2\frac{\log N}{\lambda}) \mathbf{1}_{|\mathcal{S}|} \\
&\overset{(c)}{\leq} V^\star - T V^\pi + 2\frac{\beta}{1-\gamma}(D + 2\frac{\log N}{\lambda}) \mathbf{1}_{|\mathcal{S}|}.
\end{aligned}
\tag{64}
$$

Here the inequality $(a)$ is due to the optimality of $V^\star$. The inequality $(b)$ is due to (63), and the inequality $(c)$ is due to (61). A similar derivation as (59) shows the inequality in the theorem. Hence the theorem is established. $\qquad\square$

## K    Computational Complexity of the Algorithm 1

Our overall algorithm applies a general actor-critic framework: the actor follows the proposed WPO or SPO update while the critic follows TD methods. The computational complexity depends on (i) the per-iteration computation cost of the policy and critic update and (ii) the iteration complexity of the actor-critic method. Here we mainly discuss the per-iteration computation cost of the policy update, as studies on the iteration complexity of actor-critic framework for constrained policy optimization are limited.

The computation cost of WPO and SPO updates at each iteration depends on the selection of $\beta_k$. If $\beta_k$ is chosen time dependently, the computation cost of WPO/SPO policy update is $O(n_a^2 n_s)$, where $n_a$ and $n_s$ are the number of actions and states to perform policy update. If we set $\beta_k$ as the dual optimizer, there will be additional cost to run gradient descent to solve the one-dimensional dual formulation. As discussed in our experiments, we can set $\beta_k$ to be the dual optimizer only in the first a few iterations and use a decaying afterward. Therefore, the average computational complexity of a policy update step can be $O(n_a^2 n_s)$.

## L    Difference between SPO/WPO and Other Exponential Style Updates

Sinkhorn divergence smooths the original Wasserstein by adding an entropy term, which causes the SPO update to contain exponential components similar to standard exponential style updates such as NPG (Kakade, 2001; Peng et al., 2019). Thus, SPO can be viewed as a smoother version of WPO update. Nonetheless, it's important to note that SPO/WPO updates differ fundamentally from standard exponential style updates that are based solely on entropy or KL divergence. In both SPO and WPO, the probability mass at action $a$ is redistributed to neighboring actions with high value (i.e., those $a'$ with high $A^\pi(s, a') - \beta d(a', a)$). In contrast, in these standard exponential style updates, probability mass at action $a$ is reweighted according to its exponential advantage or Q value.

## M    Exploration Properties of WPO/SPO

Compared to the Wasserstein metric, the KL divergence between policies is often larger, especially when considering the policy shifts of closely related actions, as shown in Figure 2. In practice, when employing the same trust region size $\delta$, Wasserstein metric allows for more admissible policies within the trust region compared to KL, thereby leading to better exploration. This advantage is demonstrated in our motivating example in Figure 3.

Furthermore, Sinkhorn divergence has even more exploration advantages than using Wasserstein. As Sinkhorn smooths the original Wasserstein with an entropy term, it includes additional smoother (more uniform) policies in the trust region, leading to even faster exploration.

Our numerical results in Section 7 also support that WPO/SPO explores better than KL; and SPO achieves faster exploration than WPO.

## N    Policy Parametrization, Prior Work on Nonparametric Policy

As noted in (Tessler et al., 2019), the suboptimality of policy gradient is not due to parametrization (e.g., neural network), but is a result of the parametric distribution assumption imposed on policy, which constrains policies to a predefined set. In our work, we strive to avoid suboptimality by circumventing the parametric distribution assumption imposed on policy, while still allowing for parametrization of policy in our empirical studies.

Previous research, such as (Abdolmaleki et al., 2018; Peng et al., 2019), has investigated theoretical policy update rules based on KL divergence without making explicit parametric assumptions about the policy being used. However, to our best knowledge, no prior work has explored theoretical policy update rules based on Wasserstein metric or Sinkhorn divergence.

## O  T-tests to Compare the Performance of WPO, SPO with BGPG and WNPG

We conduct independent two-sample one-tailed t-tests (Student, 1908) to compare the mean performance of our proposed methods (WPO and SPO) with two other Wasserstein-based policy optimization approaches: BGPG (Pacchiano et al., 2020) and WNPG (Moskovitz et al., 2021). Specifically, we formulate four alternative hypotheses for each task: $J_{\mathrm{WPO}} > J_{\mathrm{BGPG}}$, $J_{\mathrm{WPO}} > J_{\mathrm{WNPG}}$, $J_{\mathrm{SPO}} > J_{\mathrm{BGPG}}$, and $J_{\mathrm{SPO}} > J_{\mathrm{WNPG}}$.

MuJuCo continuous control tasks are considered for the t-tests, with a sample size of 10 for each algorithm. All t-tests are conducted at a confidence level of 90%. The results of the t-tests are presented in Table 7, where a checkmark ($\checkmark$) indicates that the alternative hypothesis is supported with 90% confidence, and a dash ($-$) indicates a failure to support the alternative hypothesis.

Based on the results presented in Table 7, we can conclude the following:

- The mean performance of WPO is higher than BGPG with 90% confidence for all tasks.

- The mean performance of WPO is higher than WNPG with 90% confidence for all tasks.

- The mean performance of SPO is higher than BGPG with 90% confidence for all tasks except Ant-v2.

- The mean performance of SPO is higher than WNPG with 90% confidence for all tasks except HalfCheetah-v2.

We note that though SPO's performance is not statistically significantly higher than BGPG or WNPG in Ant-v2 and HalfCheetah-v2 tasks, SPO demonstrates a faster convergence speed than WNPG and BGPG in these two tasks.

Table 7: T-tests results on the performance of WPO, SPO, BGPG and WNPG

| Environment | $J_{\mathrm{WPO}} > J_{\mathrm{BGPG}}$ | $J_{\mathrm{WPO}} > J_{\mathrm{WNPG}}$ | $J_{\mathrm{SPO}} > J_{\mathrm{BGPG}}$ | $J_{\mathrm{SPO}} > J_{\mathrm{WNPG}}$ |
|---|---|---|---|---|
| HalfCheetah-v2 | $\checkmark$ | $\checkmark$ | $\checkmark$ | $-$ |
| Hopper-v2 | $\checkmark$ | $\checkmark$ | $\checkmark$ | $\checkmark$ |
| Walker2d-v2 | $\checkmark$ | $\checkmark$ | $\checkmark$ | $\checkmark$ |
| Ant-v2 | $\checkmark$ | $\checkmark$ | $-$ | $\checkmark$ |
| Humanoid-v2 | $\checkmark$ | $\checkmark$ | $\checkmark$ | $\checkmark$ |

