# OpenReview forum: "Provably Convergent Policy Optimization via Metric-aware Trust Region Methods"
_TMLR — Accepted by TMLR_

### Review · Reviewer_yvPF · 2023-04-28

**Summary Of Contributions:**

This paper introduces, analyzes, and tests two conceptually simple, yet novel policy optimization algorithms based on taking gradient steps constrained by trust regions defined in terms of the Wasserstein distance (WPO) or the related Sinkhorn divergence (SPO). While previous work has studied similar ideas, this paper is the first to provide analysis of the convergence properties of such methods. Specifically, they show a performance improvement guarantee for WPO given possibly inaccurate advantage estimates, a global convergence guarantee for WPO, and that SPO converges to WPO as the regularizer coefficient $\lambda\to\infty$. Empirically, they apply practical versions of these algorithms to both tabular and discrete and continuous control tasks with function approximation, demonstrating strong performance relative to other on-policy trust region-based policy optimization algorithms.

**Audience:**

Yes

**Broader Impact Concerns:**

I don't have any concerns regarding the ethical implications of this work.

**Claims And Evidence:**

Yes

**Requested Changes:**

Requested Changes

- Empirically, results on more challenging continuous control tasks (e.g., Humanoid) seem important to support the claim that WPO/SPO are superior to other methods in this domain. Alternately, more random seeds on existing tasks would also be helpful.

- While the paper is clear overall, more discussion regarding the intuition of the WPO update would be helpful.

- A discussion surrounding $\epsilon$ in the performance improvement result for WPO would also be welcome.

- I would appreciate error bars / variance measurements be reported for the wall-clock numbers.

- Figure 10 in the Appendix seems to indicate that a KL-based method outperforms WPO for higher values of $N_A$, which doesn’t seem to be defined. Could an explanation of this result be added?

- Could the authors add runtime results for the continuous control experiments? I would think the need to perform additional sampling would drive up computational cost.

- The result regarding exploration in the motivating example at the beginning of the paper is interesting, and a discussion of how WPO/SPO can affect exploration would be very helpful.

- Could the authors add a discussion and/or results regarding the factors governing the “use optimal $\beta$ for the first 20% of runtime then decay/freeze” strategy? Would similar performance be expected for only computing the optimal $\beta$ for 10% of the run? For 5? How environment/task-dependent is this?

I am aware that all of these changes together may be tough to fit within the main paper or infeasible, but any added results or discussion in the appendix would be appreciated. Thank you!

**Strengths And Weaknesses:**

Strengths

- Overall the paper is clearly written and presented. The toy problem in Figures 1-3 is interesting and illustrative. It’s interesting that the Wasserstein distance’s sensitivity to “close” vs. “far” actions permits a *bigger* policy change in this case, which skips over the local minimum of “picking up” in the blue state. How sensitive is this phenomenon to the specific values of the geometric distances among actions, and can we think of this insight as applying to high-dimensional problem settings?

- Related work is appropriately cited and discussed.

- Most of the empirical evaluation is fairly convincing wrt the performance of both WPO and SPO, and the pros and cons of each approach is discussed nicely.

- The theoretical analysis seems thorough and rigorous, and reinforces intuition about how the proposed approaches work, though see below for more discussion.


Weaknesses

- Regarding the performance improvement guarantee, my impression is that the challenge of the exploration problem is packed away into the assumed bound on the advantage estimation error—if the max error across state-action pairs is small, then there’s an implication that the agent has strong coverage across the state(-action) space. That’s fine, but the term in the bound $-2\epsilon/(1 - \gamma)$ implies that $\epsilon$ likely has to be quite low in practice for the guarantee of improvement to hold, as $1/(1-\gamma)$ can be quite high for typical discount factors. If exploration is hard, it’s not unreasonable to expect that $\epsilon$ could be relatively large. This does seem to be a weakness of the result, although my primary concern with it is that it doesn’t seem to be discussed/addressed.

- This is relatively minor, but in several cases runtimes are reported without error bars.

- The performance on continuous control is more ambiguous—I don’t think the results support the conclusion that WPO/SPO are clearly better than other methods.

---

> ### Author Response · Authors · 2023-05-15
> **Response to Reviewer yvPF**
>
> We would like to express our appreciation to the reviewer for providing insightful comments and valuable feedback. Your input has been extremely helpful in improving the quality of our manuscript. Please find below our detailed responses to the comments and corresponding changes made to the manuscript. Thank you again for your time and effort in reviewing our work.
>
> **Q1: Empirical results on more challenging continuous control tasks (e.g., Humanoid), more random seeds**
>
> **Answer:** In our revision, we added results for the challenging continuous control task Humanoid in Figure 11 in Appendix A. We also increased the number of seeds to 10 for continuous control experiments (see Figure 8).
>
> **Q2: More discussion on the intuition of the WPO update**
>
> **Answer:** In our revision, we added a discussion on the intuition of WPO/SPO update and how WPO/SPO differs from updates based on KL or entropy in Appendix L.
>
> **Q3: Discussion surrounding $\epsilon$ in the performance improvement results (Theorem 4)**
>
> **Answer:** In our revision, we added a discussion surrounding the approximation error bound $\epsilon$ right after Theorem 4.
>
> **Q4: Error bars for runtime**
>
> **Answer:** We added the error bars for runtime in our revision. Changes are highlighted (see Table 1 and Section 7.3).
>
> **Q5: Clarification of Figure 10 in Appendix**
>
> **Answer:** $N_A$ denotes the number of samples used to estimate the advantage function. KL performs slightly better than WPO when $N_A$ is high, but WPO outperforms KL when $N_A$ decreases. This supports that compared with the KL divergence, Wasserstein metric is more robust to the inaccurate advantage estimations caused by the lack of samples. More discussion is provided in Section 7.4.
>
> **Q6: Runtime results for the continuous control experiments**
>
> **Answer:** In our revision, we added runtime results for continuous control experiments in Table 6 in Appendix A.
>
> **Q7: Discussion on how WPO/SPO can affect exploration**
>
> **Answer:** In our revision, we added
> a discussion with regards to the exploration properties of WPO/SPO in Appendix M.
>
> **Q8: The percentage of iterations with optimal $\beta$ in the "optimal-then-decay $\beta$ strategy"**
>
> **Answer:** In Appendix A.2 of our revision, we added a detailed discussion on the "optimal-then-decay $\beta$ strategy", including how we decide the number of iterations with optimal $\beta$ (we denote as $k_\beta$). We also provide $k_\beta$ value for each task in Table 3 in Appendix A.

---

> > ### Comment · Reviewer_yvPF · 2023-05-28
> > **Reviewer's Response**
> >
> > Thank you very much for making the suggested changes! I am currently leaning towards acceptance, as I believe that 1) the work appears to be correct and rigorously evaluated and 2) there are individuals in TMLR's audience who would these results of interest. I appreciate that while the theoretical results aren't particularly strong, they do align with previous results and establish a baseline guarantee of performance. I do have one final request of the authors, if possible, which is to dial down some of the claims in the paper regarding the superior empirical performance of WPO/SPO--I find the results slightly ambiguous in that regard.

---

### Review · Reviewer_dwsm · 2023-05-01

**Summary Of Contributions:**

The paper proposes to use Wasserstein distance or Sinkorn divergence as the trust region distance in policy optimization, compared to the KL divergence in the original TRPO algorithm. The paper studies the theoretical properties of such an algorithm, and provides some empirical evidence that this new trust region performs on par or better than previous baselines.

**Audience:**

Yes

**Broader Impact Concerns:**

No ethical impact.

**Claims And Evidence:**

Yes

**Requested Changes:**

I have multiple questions and comments below. The authors may consider making changes to the paper based on these questions and comments.

### Novelty of the method

It is not quite clear how much novelty this current paper brings compared to a few papers already in the literature [1,2] (both are referenced in the paper). Using metrics alternative to KL divergence is a very natural idea, and introducing distance metrics based trust region is a reasonable next step, however, this seems to have already been accomplished by prior work. If the aim here is to establish better theoretical properties / empirical performance, the results feel to me not strong enough, as I will detail below.

[1] Pacchiano et al, Learning to score behaviors for guided policy optimization, 2020
[2] Moskovitz et al, Efficient Wasserstein natural gradientsfor reinforcement learning, 2021

### Theoretical result: Thm 4

Thm 4 is a local improvement property popularized in trust region policy optimization literature. I think a major weakness here is that obtaining such a local improvement property is not very challenging since the improvement bound always contain something like $-2\epsilon/(1-\gamma)$ as shown in Eqn 10 of the paper. Such a penalty term stems from the fact that we cannot control higher order approximation error from the improvement step. It is not clear what is the benefit of such a local improvement bound compared to a KL based improvement bound. In its current form, the result does not bring much technical insight compared to existing results.

### Theoretical result: Thm 5

The global convergence property in Thm 5 is not a very strong result unfortunately. Essentially what it says is as $\lambda\rightarrow\infty$ we have $\gamma$-contraction of the value function $V^{\pi_k}$ to the optimal value function $V^\ast$. This is quite obvious since if $\lambda\rightarrow \infty$, we have $\pi_{k+1}=\arg\max A^{\pi_k}$. In other words, the algorithm reduces to policy iteration, which indeed achieves a $\gamma$-contraction. Having a more general result in Eqn 11, which considers the case when $\lambda$ and $\beta_k$ are finite, does not bring much more compared to Thm 4.

### Choice of distance metric

I think a major challenge in adopting Wasserstein distance based trust region update is that it is not clear apriori what is a good choice of the action metric $d(a_i,a_j)$. In this paper this is not discussed extensively (only that the paper suggests to implement L-1 distance throughout). Obviously, the choice of such a metric would be critical to the practical performance of the RL algorithm.

### Experiment results

The paper has studied empirical results in a few setups. Maybe the most practically relevant result is in Fig 8 where the testbeds are higher-dimensional control problems. Though it seems that there is some marginal gains of WPO/SPO compared to baseline algorithms, I do feel the performance of baseline PPO to be a bit worse than expected, compared to results reported in e.g., [3]. It is also not clear to me why the training step is generally <1M steps since such near on-policy optimization algorithms are usually benchmarked with 3-5M steps. This brings into question whether other baseline algorithms are not properly tuned with the 1M regime; or rather, it is just more convenient to run all algorithms longer for 3-5M steps and see what happens.

[3] Openai spinning up, https://spinningup.openai.com/en/latest/spinningup/bench_ppo.html

### Minor: notation

Instead of using $M$ as the distance matrix, it might be more clear to use $D$, which is compatible with the definition $D_{ij}=d(a_i,a_j)$.

**Strengths And Weaknesses:**

### Strengths

The paper provides a relatively comprehensive account of the trust region optimization algorithm using Wasserstein distance or Sinkorn divergence as the local metrics. The paper details the properties of the policy updates (e.g., convergence of Sinkorn updates to Wasserstein distance updates), local improvement properties and global convergence properties of the algorithm.

### Weaknesses

Maybe a major weakness of the paper is its lack of novelty -- the idea of using Wasserstein distance or Sinkorn divergence as the trust region metric is not new in the RL literature (see the multiple references in the paper too). The experimental results in the paper are not convincing enough to show that such a new metric leads to significantly better results than baseline KL divergence. It is also not completely clear that the theory brings much novelty and insights to the established literature.

---

> ### Author Response · Authors · 2023-05-15
> **Response to Reviewer dwsm**
>
> We would like to express our appreciation to the reviewer for providing insightful comments and valuable feedback. Your input has been extremely helpful in improving the quality of our manuscript. Please find below our detailed responses to the comments and corresponding changes made to the manuscript. Thank you again for your time and effort in reviewing our work.
>
> **Q1: Novelty of the method**
>
> **Answer:**
> While it is true that some prior work has explored alternative metrics, such as [1,2], we want to emphasize that our approach has distinct differences from these works, which we summarize below:
>
> * We consider a nonparametric policy and obtain explicit policy update forms, whereas [1,2] updates parametric policy through policy gradients.
>
> * We use Wasserstein distance to directly measure the proximity of nonparametric policies in the distribution space, whereas [1,2] measure the similarity of parametric policies in the behavioral space.
>
> * We additionally utilize Wasserstein distance as a trust region constraint, whereas [1,2] solely employ it as an explicit penalty function,
>
> In our revision, we included this discussion in the related work section on pages 3-4.
>
> [1] Pacchiano et al., Learning to score behaviors for guided policy optimization, 2020
>
> [2] Moskovitz et al., Efficient Wasserstein natural gradientsfor reinforcement learning, 2021
>
> **Q2: Theoretical result: Thm 4**
>
> **Answer:** We want to clarify that we did not claim the theoretical improvement bound of WPO (Theorem 4) is superior to that of KL. The additional term $\frac{2\epsilon}{1-\gamma}$ in WPO improvement bound is the same as the one in the improvement bound of NPG/TRPO [3]. This indicates that our methods offer comparable theoretical performance guarantees to KL based methods. However, in our experiments, we observe that the Wasserstein metric is better able to handle inaccurate advantage estimations when the sample size is small compared to KL divergence, as shown in Figure 7 and Figure 10.
>
> [3] Cen et al., Fast global convergence of natural policy gradient methods with entropy regularization, 2021
>
> **Q3: Theoretical result: Thm 5**
>
> **Answer:**  We concur with the reviewer Theorem 5 may not be as surprising given the connection with policy iteration made in [4]. However, we believe Theorem 5 is still valuable and meaningful in the following aspects:
>
> * First, this result shows that WPO and SPO
> are valid algorithms in the sense that
> they can produce an *optimal policy*, at least in the setting of infinite samples and $\beta_k \rightarrow 0$. Additionally, it provides rates for convergence speed. This feature seems to be scarce in the existing literature. Hence, we consider Theorem 5 to have theoretical value and serve as a *proof of concept* of the algorithms WPO and SPO.
> We do not need to set $\lambda \rightarrow \infty$. It can be a finite number so long as $\beta_k \rightarrow 0$.
> * Second, the theorem states that if we aim for an optimal policy, then we can set $\beta_k$ to $0$ as $k$ grows,  This aligns with what we did in the experiments. Hence, we contend that it also has some practical value.
> * Third, establishing this result is non-trivial since we are *not* performing line search on $\beta_k$ or setting $\beta_k =0$ directly.  Characterizing how a nonzero $\beta_k$ impacts the bound requires careful analysis by keeping track of the $\beta_k$ term in the derivation. As such, we believe the proof has some technical value as well.
>
>
> [4] Jalaj Bhandari and Daniel Russo, On the linear convergence of policy gradient methods for finite MDPs, 2021.
>
> **Q4: Choice of distance metric**
>
> **Answer:** Our choice of $d(a,a')$ is task-dependent and is determined through tuning. Specifically, for tasks with continuous action spaces, we use L1 distance. For tasks with discrete action space, we use 0-1 distance (i.e., $d(a,a')=0$ if $a=a'$ and $1$ otherwise). In our revision, we have included a comprehensive breakdown of our chosen distance matrices for each task in Table 3 in Appendix A for reference.

---

> > ### Author Response · Authors · 2023-05-15
> > **Response to Reviewer dwsm (Cont)**
> >
> > **Q5: Experiment results**
> >
> > **Answer:** The OpenAI Spinning Up PPO [5] results that the reviewer refers to is not the baselines implementation we use. We use TRPO, PPO and A2C from OpenAI Baselines [6] for MuJuCo continuous control experiments (Figure 8), which are benchmarked with 1M steps (see [7]). Besides, the other two Wasserstein baselines that we consider [1,2] also benchmark MuJuCo with 1M steps. Therefore, we consider 1M timesteps in our MuJuCo experiments for fair comparison. The PPO results we include in Figure 8 is consistent with OpenAI baselines results [7]. To enhance the reliability of our results, we increased the number of seeds to $10$ and added the challenging Humanoid task.
> >
> > [5] OpenAI spinning up, [https://github.com/openai/spinningup](https://github.com/openai/spinningup)
> >
> > [6] OpenAI baselines,
> > [https://github.com/openai/baselines](https://github.com/openai/baselines)
> >
> > [7] OpenAI baselines for MuJuCo,
> > [https://github.com/openai/baselines/blob/master/benchmarks_mujoco1M.htm](https://github.com/openai/baselines/blob/master/benchmarks_mujoco1M.htm)
> >
> >
> > **Q6: Notation**
> >
> > **Answer:** We renamed the cost matrix to $D$.

---

### Review · Reviewer_br9i · 2023-05-02

**Summary Of Contributions:**

This work proposes to replace Kullback-Leibler divergence in trust-region methods with Wasserstein and Sinkhorn trust regions, namely Wasserstein policy optimization (WPO) and Sinkhorn policy optimization (SPO). Algorithm 1 presents the updates for WPO/SPO.

The authors firstly use an example of grid world to intuitively compare Wasserstein and KL divergences. Figure 2 shows that Wasserstein can characterize distances between difference probability distributions over action spaces, while KL cannot. Figure 3 shows that Wasserstein-based policy update finds the optimal policy faster than KL-based policy update.

In Sections 3 and 4, the authors develop closed-form expressions of the policy updates for both WPO and SPO, using optimal Lagrangian multipliers of the trust region constraints. Results are given in Theorems 1 and 2.

Section 5 provides theoretical analysis. Theorem 4 shows that WPO has monotonic performance improvement, the same results apply for SPO with multipliers approaching infinity $\lambda \to \infty$. Theorem 5 shows that WPO has global convergence with decaying regularizer coefficient $\beta_k \to 0$.

The authors then conduct experiments to compare the proposed WPO/SPO with TRPO, PPO, A2C, and BGPG, WNPG, using several tasks, including tabular domains with discrete state and action (Taxi, Chain, and Cliff Walking), locomotion tasks with continuous state and discrete action (CartPole, Acrobot), and continuous control tasks with continuous action (HalfCheetah, Hopper, Walker, and Ant). The authors also did ablation study to examine the hyperparameter sensitivity of the proposed methods.

**Audience:**

Yes

**Broader Impact Concerns:**

The experiments are conducted using publicly available benchmarks such as Mujoco, and I do no see ethical implications of the work.

**Claims And Evidence:**

Yes

**Requested Changes:**

1.  I am wondering how computationally efficient the proposed method is comparing to using KL divergence. Maybe this could be shown by comparing the performance of different methods using the number of samples or actual running time.

2. In the motivating example, why is it the case that $1 \to 2$ in Figure 2a is a close action and $1 \to 3$ in Figure 2b is a far action?

3. Please include some closely related baselines such as SAC in the experiments.

4. It is mentioned that $\beta_k = 1/k^2$ is used in Section 7.1. Why is this specific choice and how does it compare to other possibility (e.g., $1/\sqrt{k}$, $1/k$, etc)?

**Strengths And Weaknesses:**

**Strengths**

1. Using Wasserstein and Sinkhorn in trust region policy optimization seems reasonable (from the motivating example) and new.

2. The proposed ideas are guaranteed by theory and practical algorithms.

3. The experimental results verify the proposed methods.

**Weaknesses**

1. From the closed form update of WPO (Theorem 1), it seems it is more complicated than KL divergence. This might suggest the method is computationally inefficient comparing to using KL divergence, given sometimes the proposed methods have comparable or worse performance (e.g., in Figure 6).

2. From the closed form update of SPO (Theorem 2), it looks very similar to standard exponentiate gradient update from using entropy or KL divergence. This might suggest the proposed SPO eventually is similar to existing methods, which could possibly reduce the novelty.

3. Other closely related baselines such as SAC should be included, since SAC also use entropy / KL divergence.

---

> ### Author Response · Authors · 2023-05-15
> **Response to Reviewer br9i**
>
> We would like to express our appreciation to the reviewer for providing insightful comments and valuable feedback. Your input has been extremely helpful in improving the quality of our manuscript. Please find below our detailed responses to the comments and corresponding changes made to the manuscript. Thank you again for your time and effort in reviewing our work.
>
> **Q1: Comparison of the computational efficiency of WPO and KL updates**
>
> **Answer:** We added runtime results for WPO and KL updates on five tasks from different domains to compare their computational efficiency. The runtime results for these experiments indicate that there is no notable difference in computational efficiency between WPO and KL. We reported these results in Table 5 in Appendix A in our revision.
>
> **Q2: Relationship between SPO and standard exponentiate gradient update from using entropy or KL divergence**
>
> **Answer:** Sinkhorn divergence smooths the original Wasserstein by adding an entropy term, which causes the SPO update to contain exponential components similar to standard exponential style updates such as NPG. Thus, SPO can be viewed as a smoother version of WPO update. Nonetheless, it's important to note that SPO/WPO updates differ fundamentally from standard exponential style updates that are based solely on entropy or KL divergence. In both SPO and WPO, the probability mass at action $a$ is redistributed to neighboring actions with high value (i.e., those $a'$ with high $A^\pi(s,a') - \beta d(a',a)$). In contrast,  in these standard exponential style updates, probability mass at action $a$ is reweighted according to its exponential advantage or Q value. We added this discussion in Appendix L.
>
> **Q3: Clarification of Figure 2**
>
> **Answer:** We assign a small distance of 1 between the "left" and "right" actions because they both belong to the category of directional movement and are more closely related in terms of actual control. Conversely, the "pickup" action does not fall under directional movement and is therefore assigned a larger distance to other actions (e.g., d(left,pickup)=4). We refer to policy shift $1 \xrightarrow{} 2$ as a shift of close action, since the distribution shifts from high probability "left" action to high probability "right" action, and "left" and "right" actions are relatively close based on our definition. Similarly, for policy shift $1 \xrightarrow{} 3$, the distribution shifts from high probability "left" action to high probability "pickup" action, and "left" and "pickup" actions are the furthest possible based on our definition.
>
> **Q4: SAC baseline**
>
> **Answer:** We do not consider SAC as baseline because SAC is an off-policy algorithm, whereas our WPO/SPO is on-policy. However, we include several related on-policy baselines, including TRPO, PPO (on-policy, uses KL divergence), BGPG, WNPG (on-policy, uses Wasserstein metric), etc.
>
> **Q5: Other possible decaying functions for $\beta$**
>
> **Answer:** In Appendix A.2 of our revision, we added a detailed discussion on the "optimal-then-decay $\beta$ strategy", including how we choose the decaying function $\Theta(1/\log(k))$ and how it compares to other possibilities.

---

### Review · Reviewer_xF5g · 2023-05-04

**Summary Of Contributions:**

First, I apologize to the authors for the tardiness of this review. I hope the feedback can still be useful.

This paper studies the problem of policy optimization for reinforcement learning via constraining the policy using metrics beyond the KL divergence. The paper contributes to a line of work that explores Wasserstein-based metrics for the trust-region part of policy optimization. In contrast to other works that have investigated the Wasserstein-based metrics, this work is concerned with providing theoretical guarantees and avoiding potentially suboptimal approximations or use of penalty functions. The variants of the algorithm are given, one based on the Wasserstein metric directly and one using a Sinkhorn-like version. Both involve solving their respective dual problem. A monotone improvement guarantee is established and a convergence theorem is also proven in the tabular setting.

The method is evaluated on a number of classic reinforcement learning benchmarks, covering tabular settings and continuous control settings. The proposed methods appear to outperform the baselines, although marginally.

**Audience:**

Yes

**Claims And Evidence:**

Yes

**Requested Changes:**

I have some questions and suggestions.

- I’m confused about how the cost matrix M is defined. In the motivating example, the distances are simply “assigned” by the authors. Fair enough, if you have knowledge of the problem setting in this way. However, later it goes on to say that $M_{ij} = d(a_i, a_j) = ||a_i - a_j||_1$. If these are discrete abstract actions, this would be undefined, such as in some of the experiments. How exactly do you define subtraction between two such abstract actions? In general, what would one do in the case where nothing is known about the discrete action space? How does this choice affect performance?
- How does the SPO update rule compare to NPG/SAC or other sort of “exponential weights” style algorithms for policy optimization?
It would be helpful to have more extended discussion about the policy parameterization as mentioned in the weaknesses section.
- As mentioned, the related work could be revised to include better distinction between this work and prior works rather than just discussion what prior works did.
- Potentially several more difficult control settings (e.g. Humanoid) or maybe difficult exploration problems would make the experimental results more convincing.
- Theorem 3: it would be helpful to write directly the inequality of the bound.
- Page 3: “Henceforth” used incorrectly. Consider: “Until now” or “Heretofore”
- Page 11: “coherent” used awkwardly. Consider: “consistent”

**Strengths And Weaknesses:**

*Strengths*
- Pursuing better methods for making use of the Wasserstein-based metrics is definitely an important direction.
- The derivation of the policy update rules and the theoretical results on monotone improvement and convergence appear to be new, at least for this specific problem.
- The experimental results are fairly convincing even though the improvement is marginal, but there are a few missing results that might make the case stronger.

*Weaknesses*
While I generally feel positive about the paper, there are some weaknesses that I hope can be fixed.
- The related work indeed cites a lot of the relevant literature, but falls short in properly contrasting the contributions and significance of this paper in light of that literature.
- The motivating example is a little shaky because it requires one to know in advance how the actions are related by “distance.”
- A major claim of the paper is that the update rules do not explicitly require a certain parametric class which can be restrictive (see prior work). But in practice it seems that some either implicit or explicit parameterization of the policy is used in all but the tabular settings. Theoretical update rules for KL-based methods are also do not require explicit parameterization, if I recall correctly.
- The theoretical results are not particularly concerned with exploration and generally just assume the problem away with sufficient coverage. While there is certainly more to be explored here, it may be alright considering this is on par with similar papers that study this sort of thing.
- The experimental performance improvement (although definitely there) seems marginal in practice.

---

> ### Author Response · Authors · 2023-05-15
> **Response to Reviewer xF5g**
>
> We would like to express our appreciation to the reviewer for providing insightful comments and valuable feedback. Your input has been extremely helpful in improving the quality of our manuscript. Please find below our detailed responses to the comments and corresponding changes made to the manuscript. Thank you again for your time and effort in reviewing our work.
>
> **Q1: Clarification on the choice of cost matrix**
>
> **Answer:** Our choice of $d(a,a')$ is task-dependent and is determined through tuning. Specifically, for tasks with continuous action spaces, we use L1 distance. For tasks with discrete action space, we use 0-1 distance (i.e., $d(a,a')=0$ if $a=a'$ and $1$ otherwise). In our revision, we have included a comprehensive breakdown of our chosen distance matrices for each task in Table 3 in Appendix A for reference.
>
> **Q2: SPO update rule compare to NPG/SAC or other sort of “exponential weights” style algorithms**
>
> **Answer:**
> Sinkhorn divergence smooths the original Wasserstein by adding an entropy term, which causes the SPO update to contain exponential components similar to standard exponential style updates such as NPG. Thus, SPO can be viewed as a smoother version of WPO update. Nonetheless, it's important to note that SPO/WPO updates differ fundamentally from standard exponential style updates that are based solely on entropy or KL divergence. In both SPO and WPO, the probability mass at action $a$ is redistributed to neighboring actions with high value (i.e., those $a'$ with high $A^\pi(s,a') - \beta d(a',a)$). In contrast,  in these standard exponential style updates, probability mass at action $a$ is reweighted according to its exponential advantage or Q value. We added this discussion in Appendix L.
>
> **Q3: Extended discussion on the policy parameterization**
>
> **Answer:** As noted in [1], the suboptimality of policy gradient is not due to parametrization (e.g., neural network), but is a result of the parametric distribution assumption imposed on policy, which constrains policies to a predefined set. In our work, we strive to avoid suboptimality by circumventing the parametric distribution assumption imposed on policy, while still allowing for parametrization of policy in our empirical studies.
>
> Previous research, such as [2,3], has investigated theoretical policy update rules based on KL divergence without making explicit parametric assumptions about the policy being used. However, to our best knowledge, no prior work has explored theoretical policy update rules based on Wasserstein metric or Sinkhorn divergence.
>
> We included this discussion in Appendix N in our revision.
>
> [1] Tessler et al., Distributional policy optimization: An alternative approach for continuous control, 2019
>
> [2] Abdolmaleki et al., Maximum a posteriori policy optimisation, 2018
>
> [3] Peng et al., Advantage weighted regression, 2019
>
> **Q4: Distinction between this work and previous work**
>
> **Answer:** In our revision, we updated the related work section on pages 3-4 to include more distinction between our work and previous work.
>
> **Q5: More difficult control settings (e.g. Humanoid)**
>
> **Answer:** In our revision, we added results for the challenging continuous control task Humanoid in Figure 11 in Appendix A. We also increased the number of seeds to 10 for continuous control experiment (see Figure 8).
>
> **Q6: Theorem 3: Write the inequality of the bound**
>
> **Answer:** In our revision, we added the inequality of the bound in Theorem 3.
>
> **Q7: Incorrect usages of words on page 3 and 11**
>
> **Answer:** In our revision, we corrected these words.

---

> > ### Comment · Reviewer_xF5g · 2023-06-04
> > **Thanks for the response**
> >
> > Thank you for the detailed response and updates to the paper, especially the updates to the discussion sections. This definitely clarifies many things and the new results seem promising and thorough. I've read the other reviews as well.
> >
> > Minor typo:  page 4, second paragraph "Wasserstsein"

---

### Author Response · Authors · 2023-05-15
**Summary of Our Revision**

We truly appreciate the comments from the reviewers, which greatly helped to improve the quality of the paper. We have provided detailed responses to all comments, and highlighted the changes we made in our revision. The major changes are summarized as follows:

* We added results for the challenging continuous control task Humanoid in Figure 11 in Appendix A (Reviewer xF5g, yvPF).
* We added runtime results for continuous control experiments in Table 6 in Appendix A (Reviewer yvPF).
* We increased the number of seeds to $10$ for continuous control experiments in Figure 8 (Reviewer yvPF).
* We added runtime results for WPO and KL updates in Table 5 in Appendix A (Reviewer br9i).
* We updated the related work section on pages 3-4 to include more distinction between our work and previous work (Reviewer xF5g, dwsm).
* We added a discussion on the "optimal-then-decay $\beta$ strategy" in Appendix A.2, including how we decide the number of iterations with optimal $\beta$ and the decaying function (Reviewer br9i, yvPF).
* We added a discussion on the difference between SPO and other standard exponential style updates in Appendix L (Reviewer xF5g, br9i).
* We added a discussion on the exploration properties of WPO/SPO in Appendix M (Reviewer yvPF).
* We added a discussion on policy parametrization, and the difference between our work and prior work on nonparametric policy in Appendix N (Reviewer xF5g).
* We provided our choice of distance matrix for each task in Table 3 in Appendix A (Reviewer xF5g, dwsm).
* We added error bars for wall-clock run times in Table 1 and Section 7.3 (Reviewer yvPF).

---

### Decision · Action_Editors · 2023-06-18

**Recommendation:** Accept with minor revision

**Comment:**

**Summary of the Paper:**

The paper proposes a new trust region methods for policy optimization. As opposed to TRPO, which uses the KL divergence to define the trust region, this paper suggests using the Wasserstein distance, as well as its regularized variant, to define the trust region. The resulting algorithm are WPO and SPO.
The paper derives the algorithms, shows some theoretical properties regarding their convergence, and empirically compares them with other algorithms including TRPO and two other methods (BPGD and WNPG) that also use Wasserstein distance.



**Evaluation:**

After the initial review, the authors revised their paper, and many initial concerns have been satisfactorily addressed. We have now three reviewers [br9i, xF5g, yvPF] who are Leaning Accept and one [dwsm] who is Leaning Reject.

At the high-level, the concerns are related to the novelty of the algorithm, the significant of theoretical results, and the significance of empirical results. I briefly explain them here, and postpone more discussions, and my reaction to them, to later.

*Regarding the novelty of methods:* Although there are similar algorithms (proposed by Pacchiano et al. 2020, Moskovitz et al. 2021), they are not the same. So even though the method may not be considered unprecedentedly novel, this is not a serious issue, especially because TMLR's main acceptance criteria is not based on novelty.

*Regarding theoretical result:* The theoretical results may not be the strongest. For example, the error of critic is measured in the supremum norm, which is technically simpler than an Lp norm but loses some nuances of how critic actually affects the performance. Nonetheless, I believe that is not a real issue as it still shows some basic and important properties of the algorithm.

*Regarding the empirical evidence:* The learning curves overlap significantly in some domains such as Humanoid-v2 or Ant-v2. This makes it difficult to infer much about the superiority of the algorithm, especially compared to WNPG and BGPG that use Wasserstein penalty as well. Therefore, claims such as "our methods achieve better sample efficiency, faster convergence, and improved final performance" (Item 3 in contributions in Section 1) sounds stronger than justified.


I have a few other comments from my own reading of the paper, which I will describe later. I believe they can be addressed with some work. Altogether, **I consider this paper acceptable after a few clarifications and minor revisions**.



**Detailed Comments:**

I summarize some of the comments of the reviewers, and write down my reaction to them and what I recommend the authors should do about them:

**Concern:** [yvPF,dwsm] The improvement over the baseline are not particularly convincing.

**Reaction and recommendation:**
I agree that the improvement over the baselines, especially other Wasserstein-base algorithms, are not visually significant in some domains. For example, most methods on Ant and Humanoid have a significant overlap in their shaded areas. This brings the question of what the shaded areas in the figures represent. Are they standard deviation across runs? Or standard error? Or some 95% or 95% confidence interval? (I see that in Figures 9 and 10 in the the supplementary material, standard deviation is mentioned, but not in other figures.)

My recommendation is that the authors be more clear about what they are presenting, and ideally do a statistical test to see whether the differences are actually significant or not. I know this is uncommon practice nowadays, but when the results are so close, it is warranted. It may also warrant to tone down the claim of superiority in the paper.



**Concern:** [dwsm, xF5g, yvPF] The theoretical results are not strong enough, for example, its use of the uniform error bound on the critic hides away the issue of exploration.
Moreover, at the limit of $\lambda$, they reduce to the Policy Iteration algorithm (Theorem 5). In particular, they do not shed light on why Wasserstein distance should be preferred to the KL divergence (mentioned in the private  recommendation).

**Reaction and recommendation:**
As far as I can see, the theoretical results do not explain why Wasserstein distance should be preferred, so I agree with reviewer dwsm on this front. It would be great if they did, or if they do but is not clearly apparent, the authors expand on it. This isn't a critical shortcoming for this work though.

On the other hand, I do not agree that when $\lambda$ goes to infinity, the algorithm reduces to the Policy Iteration algorithm. That would be the case when $\beta$ goes to zero, which is basically when we disregard the constraints imposed by the algorithm and "turn it off". So even if in the limit the algorithms become the same as Policy Iteration, that limit is not a good description of what the algorithm is doing and is analyzed by the theorem.

Overall, I think even though the theoretical results are perhaps not groundbreaking, they provide some evidence on the soundness of the algorithm.



**Concern:** [br9i] Comparison with SAC.

**My reaction and recommendation:** Given the similarity of the use of entropy, a comparison can be helpful, though this is optional.


**Concern:** The Wasserstein distance has been used in the policy optimization literature before (Pacchiano et al. 2020, Moskovitz et al. 2021), albeit in different forms. This somewhat reduces the novelty of the paper.

**Reaction and recommendation:**
This does not seem to be a major issue, given that there is enough differences between those algorithms and that the TMLR's criteria is not focused on the novelty anyway.

**Concern:** [xF5g, dwsm] It is not clear how to choose the action metric d.

**Reaction and recommendation:**
The question of how to choose the action metric d is an important one, but I suppose it can be done as a future work. Though I encourage the authors to explore it further for this work as well, for example, compare the effect of various choices of d on the performance.



**My Comments:**

**Major:**
* I am confused about the upper bound in Theorem 4. Can't we choose a large $\beta$ to make the RHS arbitrary large? As far as I see, all terms in the summations of the RHS are positive, so if we choose $\beta$ large enough, we show that the performance at the new iteration is arbitrary better than the performance at the old iteration. The only way I see this does not happen is if $f$ goes to zero faster than $\beta$ increases. I have not verified this myself. Please expand on this.

* One of the motivations of the using SPO over WPO has been the computational cost of computing the Wasserstein distance by solving the Optimal Transport problem compared to the Sinkhorn algorithm (for example, this is alluded in the last paragraph of Section 2). But the results in Table 5 in the supplementary material and wall-clock report in Section 7.3 do not show much of a difference. Please explain this more.



* After Theorem 2, it is mentioned that one can use gradient-based method to find the optimal solution of $\beta$. Is it established that Eq. (9) is convex?



**Minor:**

* What is distribution $\mu$ in Theorem 4? How is related to $\nu$?

* Given that most of the algorithmic development is for finite action spaces, it would be better to be upfront about that, perhaps as early as Section 1 or 2.

* Some of the references do not seem to be the right ones. For example, Mnih et al. 2013 or Silver et al. 2016 are not good examples of Policy-based RL.

* The equation numbers and references are not clickable.

* There are some typos. For example:
- Abstract and Remark 3: close-form --> closed-form
- In a few places, "few" is used instead of "a few". For example, "Recently, few work ..." (Page 1) or "In addition, few recent work ..." (Page 3).



**Requested Changes and Clarifications:**
These are the list of changes and clarification that I'd like the authors to make in their revised paper:

- Clarify the meaning of shaded areas in the revised paper and ideally perform a statistical test to show that whether the suggested methods are better than existing ones, especially for results with overlapping shaded areas. If needed, tone down the claims.
- Clarify my questions above, especially the one regarding the RHS of Eq. (10) in Theorem 4.
- Fix all the typos.
- [Optional] Compare with SAC.
- [Optional] Perform experiments with different choice of action distance d.


**Audience:**

Yes. The paper is very relevant to the Reinforcement Learning community.

**Claims And Evidence:**

Yes, most of them are supported with clear evidence. There are some minor issues that require further clarifications.

---

> ### Author Response · Authors · 2023-06-27
> **Camera Ready Version Uploaded**
>
> We would like to express our appreciation to the action editors for recommending acceptance and providing valuable feedback. Your input has been extremely helpful in improving the quality of our manuscript. Please find below our detailed responses to the comments.
>
> **Q1: Clarification of shaded areas, statistical test**
>
> **Answer:** In our camera ready revision, we have provided a clear explanation of the meaning of shaded areas in all applicable figures. Additionally, in Appendix O, we have included detailed statistical tests to compare the performance of our proposed WPO/SPO algorithm with BGPG and WNPG.
>
> **Q2: Right hand side of Eq.(10) in Theorem 4 when $\beta$ is large**
>
> **Answer:** Since $D_{ij} = 0$ if $i = j$ and $D_{ij} > 0$ if $i \ne j$, $K_{s}^{\pi_k}(\beta_k,j) = j$ holds when $\beta_k$ is large. Eq. (10) becomes ${J}(\pi_{k+1}) \ge {J}(\pi_{k}) + \beta_k E_{s \sim \rho_\upsilon^{\pi_{k+1}}} \sum_{j=1}^N \pi_{k}(a_j|s) f_s^k(j, j) D_{jj} - \frac{2\epsilon}{1-\gamma} = {J}(\pi_{k}) - \frac{2\epsilon}{1-\gamma}$. We note that the term $\beta_k E_{s \sim \rho_\upsilon^{\pi_{k+1}}} \sum_{j=1}^N \pi_{k}(a_j|s) f_s^k(j, j) D_{jj}$ vanishes due to the fact that $D_{jj} = 0$. Therefore the right hand side will not be arbitrarily large.
>
>
> **Q3: Computational cost of SPO/WPO, and results in Table 5**
>
> **Answer:** We have conducted a computational complexity analysis of our WPO/SPO algorithm, which is provided in Appendix K. The average computational complexity of a policy update step for both WPO and SPO is determined to be $O(n_a n_s^2)$, where $n_a$ and $n_s$ represent the number of actions and states involved in the policy update process. As both WPO and SPO share the same computational complexity, their wall-clock runtimes in Table $5$ do not exhibit notable differences.
>
> **Q4: Convexity of Eq.(9)**
>
> **Answer:** Eq.(9) is not convex in general. We use a global optimization algorithm called basin-hopping [1], which is able to optimize nonconvex objective.
>
> [1] Wales and Doye, Global optimization by basin-hopping and the lowest energy structures of Lennard-Jones clusters containing up to 110 atoms, 1997.
>
> **Q5: The relationship between $\mu$ and $\upsilon$**
>
> **Answer:** Both $\mu$ and $\upsilon$ denote the initial state distribution. In our camera ready revision, we unified them and used only $\upsilon$.
>
> **Q6: Upfront the fact that algorithmic development is for finite action space**
>
> **Answer:** In Section 1 of our revision (in the 2nd paragraph on page 3), we have included a discussion to emphasize the focus of our methodology and theoretical analysis on a discrete action space.
>
> **Q7: Incorrect reference, typos, equation number not clickable**
>
> **Answer:** In our camera ready revision, we updated the reference, fixed the typos and made sure the equation/reference numbers are clickable.

---

> > ### Comment · Action_Editors · 2023-07-09
> > **Thank you!**
> >
> > Thank you for the revision and your responses.